# HIERARCHICAL DEEP COUNTERFACTUAL REGRET MINIMIZATION

## ABSTRACT

Imperfect Information Games (IIGs) are used to model games under uncertainty or lack complete information. Counterfactual Regret Minimization (CFR) is one of the most successful families of algorithms for IIGs. The integration of skill-based strategy learning with CFR could potentially mirror more human-like decision-making and improve learning on complex IIGs. It enables the learning of a hierarchical strategy, wherein low-level components represent skills for solving subgames and the high-level component manages the transition between skills. In this paper, we introduce the first hierarchical version of Deep CFR (HDCFR), an innovative method that boosts learning efficiency in tasks involving extensively large state spaces and deep game trees. Notably, HDCFR enables learning with predefined (human) expertise and extracting skills transferable to similar tasks. We first present the algorithm and establish its theory in a tabular setting, including hierarchical CFR update rules and a variance-reduced Monte Carlo sampling extension for the model-free setting, where backtracking is infeasible. We then extend HDCFR to large-scale tasks via deep learning objectives that match the tabular targets under exact function fitting. Code: https://anonymous.4open.science/r/HDCFR.

## 1 INTRODUCTION

Imperfect Information Games (IIGs) can model various application domains where decision-makers have incomplete information about the state of the environment, such as auctions Noe et al. (2012), diplomacy Bakhtin et al. (2022), cybersecurity Kakkad et al. (2019). Tabular Counterfactual Regret Minimization (CFR) Zinkevich et al. (2007) has been employed in all recent milestones of Poker AI Bowling et al. (2015); Moravčík et al. (2017); Brown & Sandholm (2018), a quintessential benchmark for IIGs, highlighting the robustness of CFR. To improve scalability, researchers have proposed deep learning extensions of CFR Brown et al. (2019); Li et al. (2020); Steinberger et al. (2020), leveraging neural networks as function approximators to operate in extensive state spaces.

Professionals in a field typically possess domain-specific skills, which they compose into strategies for diverse tasks. Integrating skill-based strategy learning with CFR can enable human-like decision-making for IIGs and improve learning on complex tasks with extended decision horizons. To accomplish this, the agent learns a hierarchical strategy: low-level components represent specific skills, and the high-level component coordinates switching among skills. Notably, this is akin to the option framework Sutton et al. (1999) proposed in reinforcement learning (RL), which enables learning or planning at multiple levels of temporal abstraction. Hierarchical strategies are more interpretable, allowing humans to identify specific subgames where AI agents struggle and inject critical skills as solutions. Also, skills acquired in one task can be transferred to similar tasks to facilitate learning in new IIGs.

We propose the first hierarchical extension of Deep CFR (HDCFR), a novel approach that improves learning efficiency in tasks with deep game trees and facilitates knowledge transfer from similar games. We establish the theoretical foundations of our algorithm in the tabular setting and then introduce deep learning extensions for practical applications in large-scale games. Our contributions are as follows: **(1)** We extend the standard definition of IIGs by incorporating a hierarchical strategy and provide CFR updating rules (i.e., HCFR) for this strategy, along with convergence guarantees. **(2)** Vanilla CFR relies on a perfect game tree model and requires a complete traversal of the game tree in each training iteration, which restricts its use. Thus, we propose a sample-based model-free

extension of HCFR, including unbiased Monte Carlo estimators of counterfactual regrets and a hierarchical baseline function for variance reduction. Controlling sample variance is vital for tasks with extended decision horizons, which our algorithm targets. **(3)** We present HDCFR, where the hierarchical strategy, regret, and baseline are approximated with neural networks. The training objectives are demonstrated to be equivalent to those proposed in the tabular setting, i.e., (1) and (2), when optimality is achieved, thereby preserving the theoretical results while enjoying scalability.

We focus on the model-free outcome sampling (OS) setting, where backtracking over the full game tree is infeasible and the game model is not explicit or traversable. Accordingly, we benchmark against strong model-free OS baselines in the same setting, including DREAM, OS-MCCFR/OSSDCFR and NFSP. Our goal is to stabilize deep-horizon learning via temporal abstraction and variance reduction, without restricting the original action space.

## 2 BACKGROUND

**A detailed discussion of related works is provided in Appendix A.**

### 2.1 COUNTERFACTUAL REGRET MINIMIZATION

In an IIG Osborne & Rubinstein (1994), players make sequential moves represented by a game tree. At each non-terminal state, the player in control chooses from a set of available actions; at each terminal state, each player receives a payoff. With imperfect information, a player may not know which state they are in (e.g., in poker, a player observes private cards and public boards but not opponents' hands). Thus, each player acts based on an *information set*—a collection of indistinguishable states. Formally, the game is $\langle N, H, A, P, \sigma_c, u, \mathcal{I} \rangle$. $N$ is a finite set of players. $H$ is the set of histories, where each history is a sequence of actions from the start of the game and corresponds to a state. For $h, h' \in H$, write $h \sqsubseteq h'$ if $h$ is a prefix of $h'$. The set of actions at $h \in H$ is $A(h)$. If $a \in A(h)$, then $(ha) \in H$ is a successor history. Histories with no successors are terminal, $H_{TS} \subseteq H$. $P : H \backslash H_{TS} \to N \cup \{c\}$ maps each non-terminal history to the player in control, where $c$ is the chance player acting according to $\sigma_c(\cdot|h)$. The utility $u : N \times H_{TS} \to \mathbb{R}$ assigns a payoff to each player at every terminal history. For player $i$, $\mathcal{I}_i$ is a partition of $\{h \in H : P(h) = i\}$; each $I_i \in \mathcal{I}_i$ is an information set and represents observable information shared by all $h \in I_i$. Due to indistinguishability, $A(h) = A(I_i)$, $P(h) = P(I_i)$. Our work focuses on the two-player zero-sum setting, where $N = \{1, 2\}$ and $u_1(h) = -u_2(h)$, $\forall h \in H_{TS}$, as in prior CFR works Brown et al. (2019); Davis et al. (2020).

Every player $i \in N$ selects actions according to a strategy $\sigma_i$ that maps $I_i \in \mathcal{I}_i$ to a distribution over $A(I_i)$, and $\sigma_i(\cdot|h) = \sigma_i(\cdot|I_i)$, $\forall h \in I_i$. CFR aims to find a Nash Equilibrium (NE) strategy profile $\sigma^* = \{\sigma_1^*, \cdots, \sigma_N^*\}$, where no player has an incentive to deviate: $u_i(\sigma^*) \geq \max_{\sigma_i} u_i(\{\sigma_i, \sigma_{-i}^*\})$, $\forall i \in N$, where $-i$ denotes the players other than $i$, and $u_i(\sigma)$ is the expected payoff of player $i$:

$$u_i(\sigma) = \sum_{h' \in H_{TS}} u_i(h')\pi^\sigma(h'), \; \pi^\sigma(h') = \prod_{(ha) \sqsubseteq h'} \sigma_{P(h)}(a|I(h)) \tag{1}$$

$I(h)$ denotes the information set containing $h$, and $\pi^\sigma(h)$ is the reach probability of $h$ under $\sigma$. $\pi^\sigma(h)$ can be decomposed as $\prod_{i \in N \cup \{c\}} \pi_i^\sigma(h)$, where $\pi_i^\sigma(h) = \prod_{(ha) \sqsubseteq h', P(h)=i} \sigma_i(a|I(h))$. In addition, $\pi^\sigma(I) = \sum_{h \in I} \pi^\sigma(h)$ is the reach probability of $I$.

CFR Zinkevich et al. (2007) iteratively accumulates the counterfactual regret $R_i^T(a|I)$ for player $i$ at each information set $I \in \mathcal{I}_i$:

$$R_i^T(a|I) = \frac{1}{T} \sum_{t=1}^{T} \pi_{-i}^{\sigma^t}(I)(u_i(\sigma^t|_{I \to a}, I) - u_i(\sigma^t, I)), \; u_i(\sigma, I) = \sum_{h \in I} \pi_{-i}^\sigma(h) \sum_{h' \in H_{TS}} \pi^\sigma(h, h')u_i(h')/\pi_{-i}^\sigma(I)$$

$$\tag{2}$$

Here, $\sigma^t$ is the strategy profile at iteration $t$, $\sigma^t|_{I \to a}$ is identical to $\sigma^t$ except always choosing $a$ at $I$, and $\pi^\sigma(h, h')$ is the reach probability from $h$ to $h'$ (equals $\frac{\pi^\sigma(h')}{\pi^\sigma(h)}$ if $h \sqsubseteq h'$ and 0 otherwise). $R_i^T(a|I)$ is the expected regret of not choosing $a$ at $I$. By regret matching Abernethy et al. (2011), the next strategy $\sigma_i^{T+1}(\cdot|I)$ is defined as $\sigma_i^{T+1}(a|I) \propto \max(R_i^T(a|I), 0)$. After $T$ iterations, the

average strategy is $\overline{\sigma}_i^T(a|I) \propto \sum_{t=1}^{T} \pi_i^{\sigma^t}(I)\sigma_i^t(a|I)$. CFR guarantees that the average profile $\overline{\sigma}^T = \{\overline{\sigma}_i^T | i \in N\}$ converges to a Nash Equilibrium as $T \to \infty$.

## 2.2 THE OPTION FRAMEWORK

An option Sutton et al. (1999) $z \in \mathcal{Z}$ is specified by three components: an initiation set $Init_z \subseteq \mathcal{S}$, an intra-option policy $\sigma_z(a|s) : \mathcal{S} \times \mathcal{A} \to [0, 1]$, and a termination function $\beta_z(s) : \mathcal{S} \to [0, 1]$. $\mathcal{S}$, $\mathcal{A}$, $\mathcal{Z}$ denote the state, action, and option spaces, respectively. Option $z$ is available in state $s$ iff $s \in Init_z$. Once taken, actions follow $\sigma_z$ until stochastic termination by $\beta_z$. A high-level policy $\sigma_{\mathcal{Z}}(z|s) : \mathcal{S} \times \mathcal{Z} \to [0, 1]$ activates a new option when the previous one terminates. Thus, $\sigma_{\mathcal{Z}}(z|s)$ and $\sigma_z(a|s)$ form a hierarchical policy. Hierarchical policies tend to have superior performance on complex long-horizon tasks that can be decomposed into subtasks.

The one-step option framework Jing et al. (2021a) learns hierarchical policies without explicitly defining $Init_z$ or $\beta_z$. First, it assumes every option is available at every state, i.e., $Init_z = \mathcal{S}, \forall z \in \mathcal{Z}$. Second, it redefines the high-level and low-level policies as $\sigma^H(z \mid s, z')$ and $\sigma^L(a \mid s, z)$, respectively:

$$\sigma^H(z \mid s, z') = \beta_{z'}(s)\sigma_{\mathcal{Z}}(z \mid s) + (1 - \beta_{z'}(s))\delta_{z=z'}, \ \sigma^L(a \mid s, z) = \sigma_z(a \mid s) \tag{3}$$

where $z'$ is the option in the previous timestep and $\delta_{z=z'}$ is the indicator function. Thus, if the previous option terminates (with probability $\beta_{z'}(s)$), the agent samples a new option via $\sigma_{\mathcal{Z}}(z \mid s)$; otherwise, it continues with $z'$. The authors of Li et al. (2021a) implement the high-level policy as an end-to-end neural network with Multi-Head Attention (MHA) Vaswani et al. (2017), enabling temporal extension of options without an explicit $\beta_z$. Intuitively, if $z'$ still fits $s$, $\sigma^H(z \mid s, z')$ assigns higher attention to $z'$ and tends to continue; otherwise, it samples a more compatible option. In this one-step framework, the option is sampled at each timestep rather than only upon termination, so we train the hierarchical policy $\sigma^H$ and $\sigma^L$ directly.

## 3 METHODOLOGY

We extend CFR to learn a hierarchical strategy with neural networks (NNs) for IIGs with extensive state spaces and deep game trees. Low-level components represent skills (options) over primitive actions, while the high-level strategy coordinates their use; thus, they are learned as separate functions. Given the lack of prior hierarchical CFR, we proceed incrementally. **First,** we define the hierarchical strategy and hierarchical counterfactual regret and give CFR-style updates with a convergence guarantee. **Second,** we propose a Low-Variance Monte Carlo sampling extension to handle vast or unknown game trees—where standard traversal is impractical—without weakening the convergence rate. **Finally,** we develop HDCFR by approximating these hierarchical functions with NNs and training them via objectives that match the tabular updates under exact fitting, preserving the established theory.

## 3.1 PRELIMINARIES

In the extended game model, at each time step $t$, each player $i$ makes its decision by selecting a hierarchical action $\widetilde{a}_t \triangleq (z_t, a_t)$, which consists of the option and primitive action, based on the observable information, i.e. $I_i$. With the hierarchical actions, we can redefine the IIG model as $\langle N, H, \widetilde{A}, P, \sigma_c, u, \mathcal{I} \rangle$. Here, $N$, $P$ and $u$ retain the definitions in Section 2.1. $H$ includes all the possible histories, each of which is a sequence of hierarchical actions of all players starting from the first time step. Consequently, $\mathcal{I}$ denotes the collection of information sets induced by the new $H$, containing all observable history. $\widetilde{A}(h) = Z(h) \times A(h)$, where $Z(h)$ and $A(h)$ represent the options and primitive actions available at $h$, respectively. No action pruning: options only condition the policy and do not change $A(h)$. $\sigma_c((z_c, a)|h) = \sigma_c(a|h)$, where $\sigma_c(a|h)$ is the predefined distribution in the original game model and $z_c$ (a dummy variable) is the only option choice for the chance player.

Each player $i$ possesses a hierarchical strategy $\sigma_i(\widetilde{a}_t|I_i)$, which, by the chain rule, equals $\sigma_i^H(z_t|I_i) \cdot \sigma_i^L(a_t|I_i, z_t)$. Note that although $I_i$ includes $z_{1:t-1}$, we follow the conditional independence assumption of the one-step option framework Zhang & Whiteson (2019): $z_t \perp\!\!\!\perp z_{1:(t-2)} \mid z_{t-1}$ and $a_t \perp\!\!\!\perp z_{1:(t-1)} \mid z_t$, thus only $z_{t-1}$ ($z_t$) is used for $\sigma_i^H$ ($\sigma_i^L$) to determine $z_t$ ($a_t$). With the

hierarchical strategy, we can redefine the expected payoff and reach probability in Eq (1) by simply substituting $a$ with $\widetilde{a}$, based on which we have the definition of the average overall regret of player $i$ at iteration $T$: (From this point forward, $t$ refers to a learning iteration rather than a time step within an iteration.)

$$R_{full,i}^T = \frac{1}{T} \max_{\sigma_i'} \sum_{t=1}^{T} (u_i(\{\sigma_i', \sigma_{-i}^t\}) - u_i(\sigma^t)) \tag{4}$$

The following theorem (Theorem 2 from Zinkevich et al. (2007)) provides a connection between the average overall regret and the Nash Equilibrium solution.

**Theorem 1.** *In a two-player zero-sum game at iteration $T$, if both players' average overall regret is less than $\epsilon$, then $\overline{\sigma}^T = \{\overline{\sigma}_1^T, \overline{\sigma}_2^T\}$ is a $2\epsilon$-Nash Equilibrium.*

The average strategy $\overline{\sigma}_i^T$ is defined as Eq (5) ($\forall\, i \in N, I \in \mathcal{I}_i, \widetilde{a} \in \widetilde{A}(I)$). An $\epsilon$-Nash Equilibrium $\sigma$ approximates an NE, with the property that $u_i(\sigma) + \epsilon \geq \max_{\sigma_i'} u_i(\{\sigma_i', \sigma_{-i}\})$, $\forall\, i \in N$. Thus, $\epsilon$ measures the distance of $\sigma$ to the NE in expected payoff. Then, according to Theorem 1, as $R_{full,i}^T \to 0$ ($\forall\, i \in N$), $\overline{\sigma}^T$ converges to an NE. Theorem 1 can be applied directly to our hierarchical setting, as the only difference in derivations is the replacement of $a$ with $\widetilde{a}$ in $R_{full,i}^T$ and $\overline{\sigma}_i^T(\widetilde{a}|I)$.

$$\overline{\sigma}_i^T(\widetilde{a}|I) = \left(\Sigma_{t=1}^T \pi_i^{\sigma^t}(I)\sigma_i^t(\widetilde{a}|I)\right)/\Sigma_{t=1}^T \pi_i^{\sigma^t}(I) \tag{5}$$

## 3.2 Hierarchical CFR

A direct approach is to view $\sigma_i(\widetilde{a}|I) = \sigma_i^H(z|I) \cdot \sigma_i^L(a|I, z)$ as a unified strategy on $\widetilde{A}$ and run CFR. However, this hides the separation of levels, hindering skill reuse or human-initialized skills. We therefore introduce Hierarchical CFR (HCFR) to learn $\sigma_i^H(z|I)$ and $\sigma_i^L(a|I, z)$ separately for all $I \in \mathcal{I}_i, z \in Z(I), a \in A(I)$, and provide a convergence guarantee.

As noted in Section 3.1, achieving an NE requires minimizing the average overall regrets $R_{full,i}^T$. Instead of directly optimizing $R_{full,i}^T$, we minimize its upper bound (Theorem 2)—the sum of high-level and low-level **counterfactual regrets** at each information set: $R_i^{T,H}(z|I)$ and $R_i^{T,L}(a|I, z)$. This lets us decouple learning by independently minimizing $R_i^{T,H}(z|I)$ and $R_i^{T,L}(a|I, z)$ via $\sigma_i^H$ and $\sigma_i^L$.

**Theorem 2.** *With the following definitions of high-level and low-level counterfactual regrets:*

$$R_i^{T,H}(z|I) = \frac{1}{T} \sum_{t=1}^{T} \pi_{-i}^{\sigma^t}(I)(u_i(\sigma^t|_{I \to z}, I) - u_i(\sigma^t, I))$$

$$R_i^{T,L}(a|I, z) = \frac{1}{T} \sum_{t=1}^{T} \pi_{-i}^{\sigma^t}(I)(u_i(\sigma^t|_{Iz \to a}, Iz) - u_i(\sigma^t, Iz)) \tag{6}$$

*we have* $R_{full,i}^T \leq \sum_{I \in \mathcal{I}_i} \left[ R_{i,+}^{T,H}(I) + \sum_{z \in Z(I)} R_{i,+}^{T,L}(I, z) \right]$.

We introduce the new notations here: $\sigma^t|_{Iz \to a}$ is a hierarchical strategy profile identical to $\sigma^t$ except that the intra-option strategy of option $z$ at $I$ is always choosing $a$; $u_i(\sigma^t, Iz)$ is the expected payoff for choosing option $z$ at $I$; $R_{i,+}^{T,H}(I) = \max(\max_z R_i^{T,H}(z|I), 0)$, $R_{i,+}^{T,L}(I, z) = \max(\max_a R_i^{T,L}(a|I, z), 0)$. Proof of Theorem 2 is in Appendix B.

After obtaining the regrets $R_i^{T,H}$ and $R_i^{T,L}$, we update the high- and low-level strategies for the next iteration as follows: ($\forall\, i \in N, I \in \mathcal{I}_i, z \in Z(I), a \in A(I)$, $\mu^H$ and $\mu^L$ are normalizing factors.)

$$\sigma_i^{T+1,H}(z|I) = \begin{cases} R_{i,+}^{T,H}(z|I)/\mu^H, & \mu^H > 0, \\ 1/|Z(I)|, & o\backslash w. \end{cases}, \quad \sigma_i^{T+1,L}(a|I, z) = \begin{cases} R_{i,+}^{T,L}(a|I, z)/\mu^L, & \mu^L > 0, \\ 1/|A(I)|, & o\backslash w. \end{cases} \tag{7}$$

Thus, regrets and strategies are iteratively computed (i.e., $\sigma^{1:t} \to R^t \to \sigma^{t+1}$, $\sigma^{1:t+1} \to R^{t+1} \to \sigma^{t+2}$) with Eq (6) and (7) until convergence (i.e., $R_{full,i}^T \to 0$). The convergence rate is:

**Theorem 3.** *If player $i$ selects options and actions according to Eq (7), then $R_{full,i}^T \leq \Delta_{u,i}|\mathcal{I}_i|(\sqrt{|Z_i|} + |Z_i|\sqrt{|A_i|})/\sqrt{T}$, where $\Delta_{u,i} = \max_{h' \in H_{TS}} u_i(h') - \min_{h' \in H_{TS}} u_i(h')$, $|\mathcal{I}_i|$ is the number of information sets for player $i$, $|A_i| = \max_{h:P(h)=i} |A(h)|$, $|Z_i| = \max_{h:P(h)=i} |Z(h)|$.*

Thus, as $T \to \infty$, $R_{full,i}^T \to 0$. The rate $\mathcal{O}(T^{-0.5})$ matches CFR Zinkevich et al. (2007), so introducing options preserves convergence while enabling skill-based learning. Proof is in Appendix C.

After $T$ iterations, the average high-level and low-level strategies are: (See Appendix D for proof.)

$$\bar{\sigma}_i^{T,H}(z|I) = \frac{\Sigma_{t=1}^T \pi_i^{\sigma^t}(I)\sigma_i^{t,H}(z|I)}{\Sigma_{t=1}^T \pi_i^{\sigma^t}(I)}, \ \bar{\sigma}_i^{T,L}(a|I,z) = \frac{\Sigma_{t=1}^T \pi_i^{\sigma^t}(Iz)\sigma_i^{t,L}(a|I,z)}{\Sigma_{t=1}^T \pi_i^{\sigma^t}(Iz)} \tag{8}$$

where $\pi_i^{\sigma^t}(Iz) = \pi_i^{\sigma^t}(I)\sigma_i^{t,H}(z|I)$. Then:

**Proposition 1.** *If both players sequentially use their average high-level and low-level strategies following the one-step option model, i.e., $\forall\, I \in \mathcal{I}_i$, selecting an option $z$ according to $\bar{\sigma}_i^{T,H}(\cdot|I)$ and then selecting the action $a$ according to the corresponding intra-option strategy $\bar{\sigma}_i^{T,L}(\cdot|I,z)$, the resulting strategy profile converges to an NE as $T \to \infty$.*

### 3.3 Hierarchical Deep CFR

In vanilla CFR, regrets are updated for every information set each iteration, which is infeasible at scale. Monte Carlo CFR (MCCFR) Lanctot et al. (2009) instead updates on sampled parts of the tree. OS updates along a single sampled trajectory; ES explores all actions of the traverser and samples a single action for non-traversers, requiring an explicit traversable game model and growing exponentially with horizon. Our setting features deep trees and simulator-only access, so we adopt OS, but it suffers from high variance. **We therefore complete Section 3.2 by adding a low-variance outcome-sampling extension: replace counterfactual regrets with unbiased estimators and introduce baselines to reduce variance (Appendix E).**

Building on Section 3.2 and Appendix E, we present HDCFR, which uses NNs to approximate the regret, strategy, and baseline functions, enabling infinite-scale state spaces. **We define deep objectives that match the tabular targets under exact fitting; in practice, approximation/optimization errors may affect regret. We then present the complete algorithm in pseudo-code.** We represent each skill $z$ as a learnable embedding and implement the subpolicy $\pi_z(a \mid I)$ as a conditional network on $[I; z]$ that outputs over the full $A(I)$ (with $I$ containing both private and public observations). In particular, we train three types of networks: the counterfactual regret networks $R_{i,\theta}^{t,H}$, $R_{i,\theta}^{t,L}$, average strategy networks $\bar{\sigma}_{i,\phi}^{T,H}$, $\bar{\sigma}_{i,\phi}^{T,L}$, and baseline network $b^t$. Notably, we do not maintain the baselines $b^t$ for each player. Instead, we leverage the property of two-player zero-sum games where the payoff of the two players offsets each other. That is, $b^t = b_1^t = -b_2^t$.

**First,** the counterfactual regret networks are trained by minimizing the following two objectives: $\mathcal{L}_{R,i}^{t,H}$ and $\mathcal{L}_{R,i}^{t,L}$.

$$\mathbb{E}_{(I,\hat{r}_i^{t',H}) \sim \tau_R^i} \left[ \sum_{z \in Z(I)} (R_{i,\theta}^{t,H}(z|I) - \hat{r}_i^{t',H}(I,z))^2 \right], \ \mathbb{E}_{(Iz,\hat{r}_i^{t',L}) \sim \tau_R^i} \left[ \sum_{a \in A(I)} (R_{i,\theta}^{t,L}(a|I,z) - \hat{r}_i^{t',L}(Iz,a))^2 \right] \tag{9}$$

Here, $\tau_R^i$ stores sampled immediate counterfactual regrets $\hat{r}_i^{t'}$ from iterations 1 to $t$ (Appendix E). The core idea of MCCFR is to replace $R_i^{t,H}$ and $R_i^{t,L}$ with their unbiased Monte Carlo estimates. As a justification of this design:

**Proposition 2.** *Let $R_{i,*}^{t,H}$ and $R_{i,*}^{t,L}$ denote the minimal points of $\mathcal{L}_{R,i}^{t,H}$ and $\mathcal{L}_{R,i}^{t,L}$, respectively. For all $I \in \mathcal{I}_i$, $z \in Z(I)$, $a \in A(I)$, $R_{i,*}^{t,H}(z|I)$ and $R_{i,*}^{t,L}(a|I,z)$ yield unbiased estimations of the true counterfactual regrets scaled by positive constant factors, i.e., $C_1 R_i^{t,H}(z|I)$ and $C_2 R_i^{t,L}(a|I,z)$. Specifically, $R_{i,*}^{t,H}(z|I) \to C_1 R_i^{t,H}(z|I)$ and $R_{i,*}^{t,L}(a|I,z) \to C_2 R_i^{t,L}(a|I,z)$, as $|\tau_R^i| \to \infty$.*

Please refer to Appendix I for the proof. Since regrets are only used to compute the next strategy via Eq (7), the positive scales $C_1, C_2$ cancel in normalization; thus $R_{i,*}^{t,H}$ and $R_{i,*}^{t,L}$ can replace $R_i^{t,H}$ and $R_i^{t,L}$.

**Second,** the average strategy networks are learned from immediate strategies across iterations $1$ to $T$ by minimizing $\mathcal{L}_{\overline{\sigma},i}^{H}$ and $\mathcal{L}_{\overline{\sigma},i}^{L}$:

$$\mathbb{E}_{(I,\sigma_i^{t,H})\sim\tau_{\overline{\sigma}}^i}\left[\sum_{z\in Z(I)}(\overline{\sigma}_{i,\phi}^{T,H}(z|I)-\sigma_i^{t,H}(z|I))^2\right],\ \mathbb{E}_{(Iz,\sigma_i^{t,L})\sim\tau_{\overline{\sigma}}^i}\left[\sum_{a\in A(I)}(\overline{\sigma}_{i,\phi}^{T,L}(a|I,z)-\sigma_i^{t,L}(a|I,z))^2\right] \tag{10}$$

We adopt a sampling scheme to fulfill the following proposition. Define $q^{t,i}$ as the sample strategy profile at iteration $t$ when $i$ is the traverser; $q_p^{t,i}$ is uniformly random when $p=i$ and equals $\sigma_p^t$ when $p=3-i$. Samples in $\tau_{\overline{\sigma}}^i$ are gathered when the traverser is $3-i$ (so $i$ uses $\sigma_i^t$). Then: (See Appendix J.)

**Proposition 3.** *Let $\overline{\sigma}_{i,*}^{T,H}$ and $\overline{\sigma}_{i,*}^{T,L}$ be the minimal points of $\mathcal{L}_{\overline{\sigma},i}^{H}$ and $\mathcal{L}_{\overline{\sigma},i}^{L}$, and let $\tau_{\overline{\sigma}}^{t,i}$ be the partition of $\tau_{\overline{\sigma}}^i$ at iteration $t$. If $\tau_{\overline{\sigma}}^{t,i}$ follows the scheme above, then $\overline{\sigma}_{i,*}^{T,H}(z|I)\to\overline{\sigma}_i^{T,H}(z|I)$ and $\overline{\sigma}_{i,*}^{T,L}(a|I,z)\to\overline{\sigma}_i^{T,L}(a|I,z),\ \forall\ I\in\mathcal{I}_i,\ z\in Z(I),\ a\in A(I),\ as\ |\tau_{\overline{\sigma}}^{t,i}|\to\infty\ (t\in\{1,\cdots,T\})$.*

By Propositions 1 and 3, $\overline{\sigma}_*^{T,H}$ and $\overline{\sigma}_*^{T,L}$ can be returned as an approximate NE.

**Last,** at the end of each iteration $t$, we learn the baseline function for iteration $t+1$ to reduce variance by minimizing:

$$\mathcal{L}_b^{t+1}=\mathbb{E}_{h'\sim\tau_b^t}\left[\sum_{hza\sqsubseteq h'}(b^{t+1}(h,z,a)-\hat{b}^{t+1}(hza|h'))^2\right] \tag{11}$$

Here, $\tau_b^t$ stores trajectories collected at iteration $t$ when player 1 is the traverser. For each trajectory, we record sampled baselines $\hat{b}^{t+1}(h|h'),\forall\ h\sqsubseteq h'$, recursively defined as: ($\hat{b}^{t+1}(h|h')=u_1(h)$ if $h\in H_{TS}$)

$$\hat{b}^{t+1}(h|h')=\sum_{z\in Z(h)}\sigma_{P(h)}^{t+1,H}(z|h)\hat{b}^{t+1}(h,z|h'),\ \hat{b}^{t+1}(hz|h')=\sum_{a\in A(h)}\sigma_{P(h)}^{t+1,L}(a|h,z)\hat{b}^{t+1}(hz,a|h');$$

$$\hat{b}^{t+1}(h,z|h')=\frac{\delta(hz\sqsubseteq h')}{q^{t,1}(z|h)}\left[\hat{b}^{t+1}(hz|h')-b^t(h,z)\right]+b^t(h,z), \tag{12}$$

$$\hat{b}^{t+1}(hz,a|h')=\frac{\delta(hza\sqsubseteq h')}{q^{t,1}(a|h,z)}\left[\hat{b}^{t+1}(hza|h')-b^t(h,z,a)\right]+b^t(h,z,a).$$

The high-level baseline $b^{t+1}(h,z)$ is defined from the low-level baseline $b^{t+1}(h,z,a)$ as $b^{t+1}(h,z)=\sum_{a\in A(h)}\sigma_{P(h)}^{t+1,L}(a|I(h),z)b^{t+1}(h,z,a)$. With these sampled baselines and their relation, we have:

**Proposition 4.** *Denote $b^{t+1,*}$ as the minimal point of $\mathcal{L}_b^{t+1}$ and consider trajectories in $\tau_b^t$ as independent and identically distributed random samples. We have $b^{t+1,*}(h,z,a)\to v^{t+1,H}(\sigma^{t+1},hza)$ and $b^{t+1,*}(h,z)=\sum_{a'}\sigma_{P(h)}^{t+1,L}(a'|I(h),z)b^{t+1,*}(h,z,a')\to v^{t+1,L}(\sigma^{t+1},hz),\ \forall\ h\in H,\ z\in Z(h),\ a\in A(h),\ as\ |\tau_b^t|\to\infty.$*

Thus, the ideal criteria for baselines (Theorem 5 in Appendix E) are satisfied at the optimum of $\mathcal{L}_b^{t+1}$ (Appendix K). **In particular, the variance of estimating $R_i^t$ is minimized w.r.t. these baselines at the optimum.**

**To sum up, we present the pseudo code of HDCFR as Algorithm 1 and 2 in Appendix N.** There are $T$ iterations. **(1)** At iteration $t$, players alternate as traverser and collect $K$ trajectories (Algorithm 1, L6–11) via outcome sampling (Algorithm 2). Along each trajectory, immediate counterfactual regrets for the traverser $i$ (i.e., $\hat{r}_i^t$) are computed by Eq (26), (27) (Appendix E) and stored in $\tau_R^i$; non-traverser strategies $\sigma_{3-i}^t$ are derived from $R_{3-i,\theta}^{t-1}$ using Eq (7) and saved in $\tau_{\overline{\sigma}}^{3-i}$. **(2)** At the end of $t$, train $R_{i,\theta}^{t-1}$ on $\tau_R^i$ via Eq (9) to obtain $R_{i,\theta}^t$ (Algorithm 1, L12–14). Then update the baseline $b^t$ to $b^{t+1}$ by Eq (11) (L22–30). $b^{t+1}$ and $R_{i,\theta}^t$ are used in the next iteration. **(3)** After $T$ iterations, train $\overline{\sigma}_\phi^T$ on $\tau_{\overline{\sigma}}$ via Eq (10) (L17–19) and return it as an approximate NE.

## 4 EVALUATION AND MAIN RESULTS

In Section 4.1, we benchmark HDCFR against leading model-free methods for imperfect-information zero-sum games, including DREAM Steinberger et al. (2020), OSSDCFR (an outcome-sampling

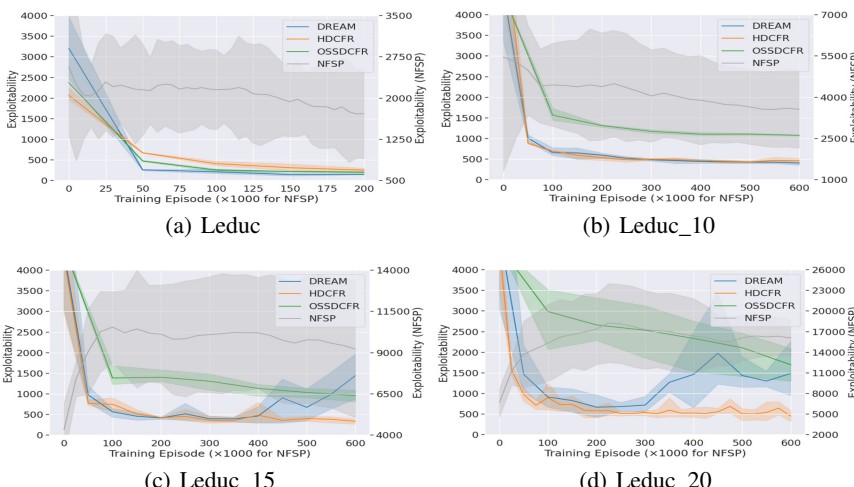

(a) Leduc                    (b) Leduc_10

(c) Leduc_15              (d) Leduc_20

Figure 1: Performance comparison on Leduc poker games. Lower exploitability indicates a closer approximation to the Nash Equilibrium. HDCFR exhibits superior convergent performance as the game's decision horizon increases; All curves are averaged over 5 random seeds

variant of Deep CFR) Steinberger (2019); Brown et al. (2019), and NFSP Heinrich & Silver (2016). Like HDCFR, these algorithms do not require domain knowledge and can be applied in environments with unknown game models (i.e., the model-free setting). For evaluation benchmarks, as a common practice, we select poker games: Leduc Southey et al. (2005) and heads-up flop hold'em (FHP) Brown et al. (2019). Given its hierarchical design, HDCFR is expected to be superior in tasks involving extended decision-making horizons. **Thus, we elevate complexity of the standard poker benchmarks by raising the number of cards and the cap on the total raises and accordingly increasing the initial stack size for each player, compelling agents to strategize over longer horizons. Details of the SIX benchmarks used are available in Table 3 in Appendix L.** Then, in Section 4.2, we analyze the hierarchical strategy learned by HDCFR. We examine whether the high-level strategy can temporally extend skills and if the low-level strategies (i.e., skills) can be transferred to new tasks to aid learning. Notably, we utilize the baseline and benchmark implementation from Steinberger (2020).

### 4.1 COMPARISON WITH STATE-OF-THE-ART MODEL-FREE ALGORITHMS ON ZERO-SUM IIGs

For Leduc poker games, we can explicitly compute the best response (BR) function for the learned strategy profile $\sigma = \{\sigma_1, \sigma_2\}$. We then can use the exploitability of $\sigma$: exploitability$(\sigma) = 1/2 \max_{\sigma'} [u_1(\sigma_1', \sigma_2) + u_2(\sigma_1, \sigma_2')]$, as the learning performance metric. Commonly used in extensive-form games, exploitability measures the deviation from an NE, so **a lower value is preferred**. For hold'em poker games (like our benchmarks), exploitability is usually quantified in milli big blinds per game (mbb/g). In Figure 1, we show the learning curves of HDCFR and the baselines. Solid lines represent the mean, while shadowed areas indicate the 95% confidence intervals from repeated trials. **(1)** For CFR-based algorithms, the players sample 900 trajectories in each training episode, and visit around $10^7$ game states in the learning process. In contrast, the RL-based NFSP algorithm is trained over more episodes ($\times 1000$) and the players visit $10^8$ game states in total during training. However, NFSP consistently underperforms across all benchmarks, so it is significantly less sample-efficient compared to the CFR-based algorithms. Note that NFSP uses a separate y-axis.

**(2)** In the absence of game models, backtracking is not allowed and so the player can sample only one action at each information set, which is known as outcome sampling. Thus, algorithms that require backtracking, like Deep CFR Brown et al. (2019) and DNCFR Li et al. (2020), cannot work directly, unless adapted with the outcome sampling scheme. However, the performance of the resulting algorithm OSSDCFR declines significantly with increasing game complexity, primarily due to the high sample variance associated with outcome sampling. **(3)** With variance reduction techniques, DREAM achieves comparable performance to HDCFR in simpler scenarios. HDCFR, owing to

Table 1: HDCFR vs. Baseline Algorithms in Head-to-Head Matchups. The table rows represent the average payoffs (higher is better) of HDCFR against the baselines: Leduc, Leduc_10, Leduc_15, Leduc_20, FHP, and FHP_10.

| Benchmark | DREAM | OSSDCFR | NFSP |
|---|---|---|---|
| Leduc | $-11.94 \pm 53.79$ | $4.11 \pm 64.03$ | $596.55 \pm 73.46$ |
| Leduc_10 | $-14.22 \pm 62.10$ | $500.0 \pm 73.22$ | $642.67 \pm 109.41$ |
| Leduc_15 | $171.33 \pm 70.80$ | $563.75 \pm 83.31$ | $1351.5 \pm 207.27$ |
| Leduc_20 | $196.89 \pm 76.69$ | $587.0 \pm 68.83$ | $1725.33 \pm 206.01$ |
| FHP | $184.58 \pm 36.75$ | $68.11 \pm 36.61$ | $244.61 \pm 41.36$ |
| FHP_10 | $282.42 \pm 14.20$ | $343.22 \pm 15.35$ | $537.39 \pm 16.91$ |

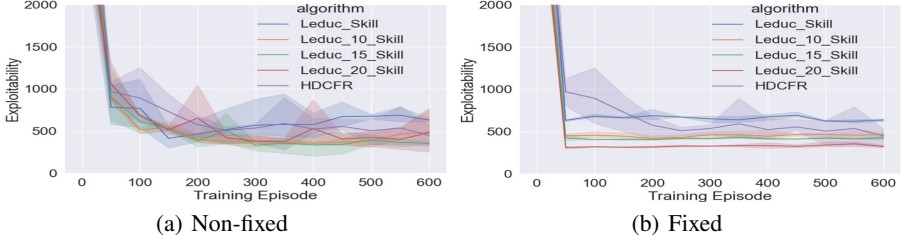

(a) Non-fixed         (b) Fixed

Figure 2: Learning performance on Leduc_20 with transferred skills from other Leduc tasks. The transferred skills can either be fixed or not when learning a hierarchical strategy on the new scenario. As a reference, the learning performance without transferred skills is labeled as HDCFR. By fixing pre-learned skills, the agent focuses on mastering a high-level strategy, thus accelerating learning. However, adjusting these skills alongside the high-level strategy can yield improved results: when using Leduc_15 skills, peaked performance occurs around episode 400.

its hierarchical structure, excels over DREAM in games with extended horizons, where DREAM struggles to converge. HDCFR's superiority becomes more significant as the game complexity increases.On Leduc_15/20, DREAM's exploitability occasionally decreases and then increases—this occurs in long-horizon settings where outcome sampling has high variance. Our ablations (Appendix M, Fig. 3a) reproduce a similar pattern when removing the baseline or the multi-head attention module (no temporally extended skills), indicating that variance control and temporal abstraction are crucial for stability in deep horizons.

Further, we conduct head-to-head tournaments between HDCFR and each baseline. We select the top three checkpoints for each algorithm, resulting in total nine pairings for each tournament. Each pair of strategies competes over 1,000 hands. Table 1 shows the average payoff of HDCFR's strategy profile (calculated as $1/2 \left[ u_1(\sigma_1^{\text{HDCFR}}, \sigma_2^{\text{baseline}}) + u_2(\sigma_1^{\text{baseline}}, \sigma_2^{\text{HDCFR}}) \right]$), along with 95% confidence intervals, measured in mbb/g. **A higher payoff indicates superior decision-making performance.** Observations from Leduc poker games in this table (i.e., rows 1-4) align with aforementioned conclusions (1)-(3). **To further show the superiority of our algorithm, we compare it with baselines on larger-scale FHP games, which boast a game tree exceeding $10^{12}$ nodes in size.** Due to the immense scale of FHP games, computing best response functions is impractical, so we offer only head-to-head comparison results. Training an instance on FHP games requires 7 days using a device with 8 CPU cores (3rd Gen Intel Xeon) and 128 GB of RAM. We utilized the RAY parallel computing framework Moritz et al. (2018). We can see that HDCFR's advantage grows as task difficulty goes up.

## 4.2 ANALYSIS ON THE LEARNED HIERARCHICAL STRATEGY

One key benefit of hierarchical learning is to use prelearned skills as building blocks for strategy learning, providing a manner for integrating expert knowledge. Even in the absence of domain knowledge, where rule-based skills can't be provided as expert guidance, we can leverage skills learned from similar games. Skills, as policy segments, often exhibit greater transferability than complete strategies. In Figure 2, we demonstrate the transfer of skills from various Leduc games to Leduc_20 and present the learning outcomes. For comparison, we also include the performance

Table 2: Comparison of the switch frequencies when using skills from different source tasks.

| Source Task | Leduc | Leduc_10 | Leduc_15 | Leduc_20 |
|---|---|---|---|---|
| Switch Frequency | 0.1363 $\pm\, 4.02 \times 10^{-4}$ | 0.1256 $\pm\, 3.43 \times 10^{-4}$ | 0.1088 $\pm\, 2.16 \times 10^{-4}$ | 0.1016 $\pm\, 3.64 \times 10^{-4}$ |

without transferred skills, labeled as HDCFR. We define skill-switch frequency as the ratio between the number of time steps at which the selected option changes and the total number of time steps; its inverse approximates the average option duration. These prelearned skills can either remain static (Figure 2(b)) or be trained alongside the high-level strategy (Figure 2(a)). When kept static, the agent can focus on mastering its high-level strategy to select among a set of effective skills, resulting in quicker convergence. Notably, the superior convergent performance observed in Figure 2(b) positively correlates with the similarity between the source task of the skills and Leduc_20. On the other hand, if the skills are adjusted with the high-level strategy, the improvement on the convergence speed may not be obvious, but skills can be more customized for the current task and better performance may be achieved. For instance, with Leduc_15 skills, peak performance is achieved around episode 400; for Leduc skills, training with dynamic skills (Figure 2(a)) results in better performance compared to static skills (Figure 2(b)). However, for Leduc_20 skills, fixed skills works better. This could be because they originate from the same task, eliminating the need for further adaptation.

Next, we provide an analysis of the learned high-level strategies. As depicted in Figure 2(b), when utilizing fixed skills from different source tasks, corresponding high-level strategies can be learned. To evaluate whether these high-level strategies can extend skills temporally (with the attention mechanism) – instead of frequently toggling between skills – we calculate the frequency of skill switches in the game tree of Leduc_20 (containing 113954 nodes in total), considering all possible hands of cards and five repeated experiments. Table 2 reports the means and 95% confidence intervals of the skill switch frequency. As the decision horizon of the skills' source task increases, the switch frequency decreases, reflecting longer durations of single-skill utilization. Notably, for Leduc_20 skills, skill switches occur only about 10% of the time. This indicates the agent's preference for decision-making at an extended-skill level, with an average skill duration of approximately 10 steps, rather than at the level of primitive actions, aligning with our expectations. **An ablation study is provided in Appendix M to highlight the importance of each component within our algorithm.**

## 5 CONCLUSION

We introduce the first hierarchical extension of CFR based on the option framework. We first establish the theoretical foundations in a tabular setting and then extend them using neural networks as function approximators, resulting in a theoretically grounded deep learning algorithm – HDCFR. Evaluations in complex two-player zero-sum games show that HDCFR outperforms leading algorithms in this field, with the advantage increasing as the decision horizon grows, underscoring its potential for tasks involving deep game trees. Moreover, we show that the learned high-level strategy can *temporally* extend skills to exploit hierarchical subtask structure in long-horizon tasks, and that the learned skills can be transferred to related tasks to facilitate learning. Our algorithm provides a novel framework to learn with pre-learned skills in zero-sum IIGs.

**Ethics Statement**    This work complies with the ICLR Code of Ethics. While our methods are general, they may be applied in contexts with societal implications, including risks related to bias, fairness, and privacy. We encourage responsible use and declare no conflicts of interest.

**Reproducibility Statement**    We provide detailed descriptions of our methodology, datasets, model configurations, and evaluation metrics in both the main text and the Appendix. In addition, the complete source code is included in the supplementary materials to facilitate reproducibility.

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

## A RELATED WORKS

Counterfactual Regret Minimization (CFR) Zinkevich et al. (2007) is an algorithm for learning Nash Equilibria in extensive-form games through iterative self-play. As part of this process, it must traverse the entire game tree during each learning iteration, which is prohibitive for large-scale games. This motivates the development of Monte Carlo CFR (MCCFR) Lanctot et al. (2009), which samples trajectories traversing part of the tree to allow for significantly faster iterations. However, the variance of Monte Carlo sampling can become a significant issue, particularly for long sample trajectories. The authors of Schmid et al. (2019); Davis et al. (2020) then propose to introduce baseline functions for variance reduction. Notably, all methods mentioned above are tabular-based. For games with large state space, domain-specific abstraction schemes Ganzfried & Sandholm (2014); Moravčík et al. (2017) are required to shrink them to a manageable size by clustering states into buckets, which necessitates expert knowledge and is not applicable to all games.

To obviate the need of abstractions, several CFR variants utilizing function approximators have been proposed. Pioneering this was Regression CFR Waugh et al. (2015), which adopts regression trees to model cumulative regrets but relies on hand-crafted features and full traversals of the game tree. Subsequently, several works Brown et al. (2019); Li et al. (2020); Steinberger (2019); Li et al. (2021b) propose to model the cumulative counterfactual regrets and average strategies in MCCFR as neural networks to enhance the scalability. **However, all these methods rely on knowledge of the game model to realize backtracking (i.e., sampling multiple actions at an information set) for regret estimation.** As a model-free approach, Neural Fictitious Self-Play (NFSP) Heinrich & Silver (2016) is the first deep reinforcement learning algorithm capable of learning a Nash Equilibrium in two-player imperfect information games through self-play. Since its introduction, various policy gradient and actor-critic methods have demonstrated similar convergence properties when appropriately tuned Lanctot et al. (2017); Srinivasan et al. (2018). However, fictitious play empirically converges slower than CFR-based approaches in many settings. DREAM Steinberger et al. (2020) extends Deep CFR with variance-reduction techniques from Davis et al. (2020) and represents the state-of-the-art in model-free algorithms of this area. **Compared with DREAM, our algorithm provides solid theoretical justification, enables hierarchical learning with (prelearned) skills, and empirically shows enhanced performance on longer-horizon games.**

As another important module of HDCFR, the option framework Sutton et al. (1999) enables learning and planning at multiple temporal levels and has been widely adopted in reinforcement learning (RL). Multiple research areas centered on this framework have been developed. Unsupervised Option Discovery focuses on identifying skills that are diverse and effective for (various) downstream tasks without relying on task-specific reward signals. Algorithms have been developed for both **single-agent settings** Eysenbach et al. (2019); Jinnai et al. (2020); Chen et al. (2022a) and **collaborative multi-agent scenarios** Chen et al. (2022c;b); Zhang et al. (2022). Hierarchical Reinforcement Learning Zhang & Whiteson (2019); Li et al. (2021a) and Hierarchical Imitation Learning Jing et al. (2021b); Chen et al. (2023a;b), on the other hand, aim at directly learning a hierarchical policy that incorporates skills, either through interactions with the environment or from expert demonstrations.

Our proposed algorithm – HDCFR, is a pioneering effort to amalgamate options with CFR and demonstrates the superiority of hierarchical learning in **competitive multi-agent scenarios** (more specifically, zero-sum games).

## B PROOF OF THEOREM 2

Define $D(I)$ to be the information sets of player $i$ reachable from $I$ (including $I$), and $\sigma|_{D(I) \to \sigma_i'}$ to be a strategy profile equal to $\sigma$ except that player $i$ adopts $\sigma_i'$ in the information sets contained in $D(I)$. Then, the average overall regret starting from $I$ ($I \in \mathcal{I}_i$) can be defined as:

$$R_{full,i}^T(I) = \frac{1}{T} \max_{\sigma_i'} \sum_{t=1}^{T} \pi_{-i}^{\sigma^t}(I)(u_i(\sigma^t|_{D(I) \to \sigma_i'}, I) - u_i(\sigma^t, I)) \tag{13}$$

Further, we define $S_i(I, \widetilde{a})$ to be the set of all possible next information sets of player $i$ given that action $\widetilde{a} \in \widetilde{A}(I)$ was just selected at $I$ and define $S_i(I) = \bigcup_{\widetilde{a} \in \widetilde{A}(I)} S_i(I, \widetilde{a})$, $S_i(Iz) = \bigcup_{a \in A(I)} S_i(I, za)$. Then, we have the following lemma:

**Lemma 1.** $R_{full,i}^{T,+}(I) \leq R_{i,+}^{T,H}(I) + \sum_{z \in Z(I)} R_{i,+}^{T,L}(I,z) + \sum_{I' \in S_i(I)} R_{full,i}^{T,+}(I')$

*Proof.*

$$R_{full,i}^T(I) = \frac{1}{T} \max_{\sigma_i'} \sum_{t=1}^T \pi_{-i}^{\sigma^t}(I)(u_i(\sigma^t|_{D(I) \to \sigma_i'}, I) - u_i(\sigma^t, I))$$

$$= \frac{1}{T} \max_{z \in Z(I)} \max_{\sigma_i'} \sum_{t=1}^T \pi_{-i}^{\sigma^t}(I) \left[ (u_i(\sigma^t|_{I \to z}, I) - u_i(\sigma^t, I)) + (u_i(\sigma^t|_{D(Iz) \to \sigma_i'}, Iz) - u_i(\sigma^t, Iz)) \right]$$

$$\leq \frac{1}{T} \max_{z \in Z(I)} \sum_{t=1}^T \pi_{-i}^{\sigma^t}(I)(u_i(\sigma^t|_{I \to z}, I) - u_i(\sigma^t, I)) + \frac{1}{T} \max_{z \in Z(I)} \max_{\sigma_i'} \sum_{t=1}^T \pi_{-i}^{\sigma^t}(I)(u_i(\sigma^t|_{D(Iz) \to \sigma_i'}, Iz) - u_i(\sigma^t, Iz))$$

$$= R_i^{T,H}(I) + \frac{1}{T} \max_{z \in Z(I)} \max_{\sigma_i'} \sum_{t=1}^T \pi_{-i}^{\sigma^t}(Iz)(u_i(\sigma^t|_{D(Iz) \to \sigma_i'}, Iz) - u_i(\sigma^t, Iz))$$

$$\leq R_i^{T,H}(I) + \sum_{z \in Z(I)} \left[ \frac{1}{T} \max_{\sigma_i'} \sum_{t=1}^T \pi_{-i}^{\sigma^t}(Iz)(u_i(\sigma^t|_{D(Iz) \to \sigma_i'}, Iz) - u_i(\sigma^t, Iz)) \right]^+$$

$$= R_i^{T,H}(I) + \sum_{z \in Z(I)} R_{full,i}^{T,+}(Iz) \tag{14}$$

$$R_{full,i}^T(Iz) = \frac{1}{T} \max_{\sigma_i'} \sum_{t=1}^T \pi_{-i}^{\sigma^t}(Iz)(u_i(\sigma^t|_{D(Iz) \to \sigma_i'}, Iz) - u_i(\sigma^t, Iz))$$

$$= \frac{1}{T} \max_{a \in A(I)} \max_{\sigma_i'} \sum_{t=1}^T \pi_{-i}^{\sigma^t}(Iz)[(u_i(\sigma^t|_{Iz \to a}, Iz) - u_i(\sigma^t, Iz)) +$$

$$\sum_{I' \in S_i(I,za)} P_{\sigma_{-i}^t}(I'|I, za)(u_i(\sigma^t|_{D(I') \to \sigma_i'}, I') - u_i(\sigma^t, I'))] \tag{15}$$

$$= R_i^{T,L}(I,z) + \max_{a \in A(I)} \sum_{I' \in S_i(I,za)} \frac{1}{T} \max_{\sigma_i'} \sum_{t=1}^T \pi_{-i}^{\sigma^t}(I')(u_i(\sigma^t|_{D(I') \to \sigma_i'}, I') - u_i(\sigma^t, I'))$$

$$= R_i^{T,L}(I,z) + \max_{a \in A(I)} \sum_{I' \in S_i(I,za)} R_{full,i}^T(I')$$

$$\leq R_i^{T,L}(I,z) + \sum_{I' \in S_i(Iz)} R_{full,i}^{T,+}(I')$$

In Equation (14) and (15), we employ the one-step look-ahead expansion (Equation (10) in Zinkevich et al. (2007)) for the second line. At iteration $t$, when player $i$ selects a hierarchical action $\tilde{a} = (za)$, it will transit to the subsequent information set $I' \in S_i(I, za)$ with a probability of $P_{\sigma_{-i}^t}(I'|I, za)$, since only player $-i$ will act between $I$ and $I'$ according to $\sigma_{-i}^t$. According to the definition of the reach probability, $\pi_{-i}(Iz) = \pi_{-i}(I)$ (since $z$ and $a$ are executed by player $i$) and $\pi_{-i}(I)P_{\sigma_{-i}^t}(I'|I, za) = \pi_{-i}(I')$. Combining Equation (14) and (15), we can get:

$$R_{full,i}^T(I) \leq R_i^{T,H}(I) + \sum_{z \in Z(I)} R_{full,i}^{T,+}(Iz)$$

$$\leq R_i^{T,H}(I) + \sum_{z \in Z(I)} \left[ R_{i,+}^{T,L}(I,z) + \sum_{I' \in S_i(Iz)} R_{full,i}^{T,+}(I') \right] \tag{16}$$

$$= R_i^{T,H}(I) + \sum_{z \in Z(I)} R_{i,+}^{T,L}(I,z) + \sum_{z \in Z(I)} \sum_{I' \in S_i(Iz)} R_{full,i}^{T,+}(I')$$

$$= R_i^{T,H}(I) + \sum_{z \in Z(I)} R_{i,+}^{T,L}(I,z) + \sum_{I' \in S_i(I)} R_{full,i}^{T,+}(I')$$

In previous derivations, we have repeatedly employed the inequality $\max(a + b, 0) \leq \max(a, 0) + \max(b, 0)$, which holds for all $a, b \in \mathbb{R}$, as in the last inequality of Equation (14) and (15). By applying this inequality once more to Equation (16), we can obtain Lemma 1. $\qquad \square$

**Lemma 2.** $R_{full,i}^{T,+}(I) \leq \sum_{I' \in D(I)} \left[ R_{i,+}^{T,H}(I') + \sum_{z \in Z(I')} R_{i,+}^{T,L}(I', z) \right]$

*Proof.* We prove this lemma by induction on the height of the information set $I$ on the game tree. When the height is 1, i.e., $S_i(I) = \emptyset$, $D(I) = \{I\}$, then Lemma 1 implies Lemma 2. Now, for the general case:

$$R_{full,i}^{T,+}(I) \leq R_{i,+}^{T,H}(I) + \sum_{z \in Z(I)} R_{i,+}^{T,L}(I, z) + \sum_{I' \in S_i(I)} R_{full,i}^{T,+}(I')$$

$$\leq R_{i,+}^{T,H}(I) + \sum_{z \in Z(I)} R_{i,+}^{T,L}(I, z) + \sum_{I' \in S_i(I)} \sum_{I'' \in D(I')} \left[ R_{i,+}^{T,H}(I'') + \sum_{z \in Z(I'')} R_{i,+}^{T,L}(I'', z) \right] \quad (17)$$

$$= \sum_{I' \in D(I)} \left[ R_{i,+}^{T,H}(I') + \sum_{z \in Z(I')} R_{i,+}^{T,L}(I', z) \right]$$

In the second line, we employ the induction hypothesis. In the third line, we use the following facts: $D(I) = \{I\} \cup \bigcup_{I' \in S_i(I)} D(I')$, $\{I\} \cap \bigcup_{I' \in S_i(I)} D(I') = \emptyset$, and $D(I') \cap D(I'') = \emptyset$ for all distinct $I', I'' \in S_i(I)$. The third fact here is derived from the perfect recall property of the game: all players can recall their previous (hierarchical) actions and the corresponding information sets. Then, $D(I') \cap D(I'') = \emptyset$ because elements from the two sets possess distinct prefixes (i.e., $I'$ and $I''$). $\qquad \square$

Last, for the average overall regret, we have $R_{full,i}^T = R_{full,i}^T(\emptyset)$, where $\emptyset$ corresponds to the start of the game tree and $D(\emptyset) = \mathcal{I}_i$. Applying Lemma 2, we can get the theorem: $R_{full,i}^T \leq R_{full,i}^{T,+}(\emptyset) \leq \sum_{I \in \mathcal{I}_i} \left[ R_{i,+}^{T,H}(I) + \sum_{z \in Z(I)} R_{i,+}^{T,L}(I, z) \right]$.

## C    PROOF OF THEOREM 3

Regret matching can be defined in a domain where a fixed set of actions $A$ and a payoff function $u^t : A \rightarrow \mathbb{R}$ exist. At each iteration $t$, a distribution over the actions, $\sigma^t$, is chosen based on the cumulative regret $R^t : A \rightarrow \mathbb{R}$. Specifically, the cumulative regret at iteration $T$ for not playing action $a$ is defined as:

$$R^T(a) = \frac{1}{T} \sum_{t=1}^{T} \left[ u^t(a) - \sum_{a' \in A} \sigma^t(a') u^t(a') \right] \quad (18)$$

where $\sigma^t(a)$ is obtained by:

$$\sigma^t(a) = \begin{cases} R^{t-1,+}(a)/\mu, & \mu > 0, \\ 1/|A|, & o \backslash w. \end{cases} \quad \mu = \sum_{a' \in A} R^{t-1,+}(a') \quad (19)$$

Then, we have the following lemma (Theorem 8 in Zinkevich et al. (2007)):

**Lemma 3.** $\max_{a \in A} R^T(a) \leq \frac{\Delta_u \sqrt{|A|}}{\sqrt{T}}$, *where* $\Delta_u = \max_{t \in \{1, \cdots, T\}} \max_{a,a' \in A} (u^t(a) - u^t(a'))$.

To apply this lemma, we must transform the definitions of $R_i^{T,H}$ and $R_i^{T,L}$ in Equation (6) to a form resembling Equation (18). With Equation (6) and (2), we can get:

$$
\begin{aligned}
R_i^{T,H}(z|I) &= \frac{1}{T} \sum_{t=1}^{T} \left[ \sum_{h \in I} \pi_{-i}^{\sigma^t}(h) \sum_{h' \in H_{TS}} u_i(h') \pi^{\sigma^t}(hz, h') - \right. \\
&\qquad \left. \sum_{h \in I} \pi_{-i}^{\sigma^t}(h) \sum_{h' \in H_{TS}} u_i(h') \sum_{z' \in Z(h)} \sigma_t^H(z'|h) \pi^{\sigma^t}(hz', h') \right] \\
&= \frac{1}{T} \sum_{t=1}^{T} \left[ \sum_{h \in I} \pi_{-i}^{\sigma^t}(h) \sum_{h' \in H_{TS}} u_i(h') \pi^{\sigma^t}(hz, h') - \right. \\
&\qquad \left. \sum_{z' \in Z(I)} \sigma_t^H(z'|I) \sum_{h \in I} \pi_{-i}^{\sigma^t}(h) \sum_{h' \in H_{TS}} u_i(h') \pi^{\sigma^t}(hz', h') \right] \\
&= \frac{1}{T} \sum_{t=1}^{T} \left[ v_t^H(z) - \sum_{z' \in Z(I)} \sigma_t^H(z'|I) v_t^H(z') \right]
\end{aligned}
\tag{20}
$$

Applying the same process on $R_i^{T,L}(a|I, z)$, we can get:

$$
\begin{aligned}
R_i^{T,L}(a|I, z) &= \frac{1}{T} \sum_{t=1}^{T} \left[ v_t^L(a) - \sum_{a' \in A(I)} \sigma_t^L(a'|I, z) v_t^L(a') \right] \\
v_t^L(a) &= \sum_{h \in I} \pi_{-i}^{\sigma^t}(h) \sum_{h' \in H_{TS}} u_i(h') \pi^{\sigma^t}(hza, h')
\end{aligned}
\tag{21}
$$

Then, we can apply Lemma 3 and obtain:

$$
\begin{aligned}
\max_{z \in Z(I)} R_i^{T,H}(z|I) = R_i^{T,H}(I) &\leq \frac{\Delta_{v^H} \sqrt{|Z(I)|}}{\sqrt{T}} \leq \frac{\Delta_{u,i} \sqrt{|Z(I)|}}{\sqrt{T}} \\
\max_{a \in A(I)} R_i^{T,L}(a|I, z) = R_i^{T,L}(I, z) &\leq \frac{\Delta_{v^L} \sqrt{|A(I)|}}{\sqrt{T}} \leq \frac{\Delta_{u,i} \sqrt{|A(I)|}}{\sqrt{T}}
\end{aligned}
\tag{22}
$$

Here, $\Delta_{u,i} = \max_{h' \in H_{TS}} u_i(h') - \min_{h' \in H_{TS}} u_i(h')$ is the range of the payoff function for $i$, which covers $\Delta_{v^H}$ and $\Delta_{v^L}$. We can directly apply Lemma 3, because the regret matching is adopted at each information set independently as defined in Equation (7). By integrating Equation 22 and Theorem 2, we then get:

$$
\begin{aligned}
R_{full,i}^T &\leq \sum_{I \in \mathcal{I}_i} \left[ \frac{\Delta_{u,i} \sqrt{|Z(I)|}}{\sqrt{T}} + \sum_{z \in Z(I)} \frac{\Delta_{u,i} \sqrt{|A(I)|}}{\sqrt{T}} \right] \\
&\leq \frac{\Delta_{u,i} |\mathcal{I}_i|}{\sqrt{T}} (\sqrt{|Z_i|} + |Z_i| \sqrt{|A_i|})
\end{aligned}
\tag{23}
$$

where $|\mathcal{I}_i|$ is the number of information sets for player $i$, $|A_i| = \max_{h:P(h)=i} |A(h)|$, $|Z_i| = \max_{h:P(h)=i} |Z(h)|$.

# D    PROOF OF PROPOSITION 1

According to Theorem 1 and 3, as $T \to \infty$, $R_{full,i}^T \to 0$, and thus the average strategy $\overline{\sigma}_i^T(\widetilde{a}|I)$ converges to a Nash Equilibrium. We claim that $\overline{\sigma}_i^T(\widetilde{a}|I) = \overline{\sigma}_i^{T,H}(z|I) \cdot \overline{\sigma}_i^{T,L}(a|I, z)$.

*Proof.*

$$\overline{\sigma}_i^{T,H}(z|I) \cdot \overline{\sigma}_i^{T,L}(a|I,z) = \frac{\Sigma_{t=1}^T \pi_i^{\sigma^t}(I)\sigma_i^{t,H}(z|I)}{\Sigma_{t=1}^T \pi_i^{\sigma^t}(I)} \frac{\Sigma_{t=1}^T \pi_i^{\sigma^t}(Iz)\sigma_i^{t,L}(a|I,z)}{\Sigma_{t=1}^T \pi_i^{\sigma^t}(Iz)}$$

$$= \frac{\Sigma_{t=1}^T \pi_i^{\sigma^t}(I)\sigma_i^{t,H}(z|I)}{\Sigma_{t=1}^T \pi_i^{\sigma^t}(I)} \frac{\Sigma_{t=1}^T \pi_i^{\sigma^t}(Iz)\sigma_i^{t,L}(a|I,z)}{\Sigma_{t=1}^T \pi_i^{\sigma^t}(I)\sigma_i^{t,H}(z|I)} \tag{24}$$

$$= \frac{\Sigma_{t=1}^T \pi_i^{\sigma^t}(Iz)\sigma_i^{t,L}(a|I,z)}{\Sigma_{t=1}^T \pi_i^{\sigma^t}(I)} = \frac{\Sigma_{t=1}^T \pi_i^{\sigma^t}(I)\sigma_i^{t,H}(z|I)\sigma_i^{t,L}(a|I,z)}{\Sigma_{t=1}^T \pi_i^{\sigma^t}(I)}$$

$$= \frac{\Sigma_{t=1}^T \pi_i^{\sigma^t}(I)\sigma_i^t((z,a)|I)}{\Sigma_{t=1}^T \pi_i^{\sigma^t}(I)} = \overline{\sigma}_i^T(\widetilde{a}|I)$$

$\square$

Given this equivalence, we can infer that if both players adhere to the one-step option model for each $I$ – selecting an option $z$ based on $\overline{\sigma}_i^{T,H}(\cdot|I)$ and subsequently choosing the action $a$ in accordance with the corresponding intra-option strategy $\overline{\sigma}_i^{T,L}(\cdot|I,z)$, an approximate NE solution can be acquired.

## E  LOW-VARIANCE MONTE CARLO SAMPLING EXTENSION

MCCFR's main insight is substituting the counterfactual regrets $R_i^T$ in CFR with their unbiased estimations, while maintaining the other learning rules (as in Section 3.2). This approach allows for updating functions only at information sets within the sample trajectories, bypassing the need to traverse the full game tree. With MCCFR, the average overall regret $R_{full,i}^T \to 0$ as $T \to \infty$ at the same convergence rate as vanilla CFR, with high probability, as stated in Theorem 5 of Lanctot et al. (2009). Therefore, to apply the Monte Carlo sampling extension, we propose unbiased estimations of $R_i^{T,H}(z|I)$ and $R_i^{T,L}(a|I,z), \forall\, i \in N, I \in \mathcal{I}_i, z \in Z(I), a \in A(I)$.

First, we define $R_i^{T,H}(z|I)$ and $R_i^{T,L}(a|I,z)$ with the immediate counterfactual regrets $r_i^t$ and values $v_i^t$: $(v_i^{t,H}(\sigma^t, h) = u_i(h), \forall\, h \in H_{TS})$

$$R_i^{T,H}(z|I) = \frac{1}{T}\sum_{t=1}^T r_i^{t,H}(I,z), \; r_i^{t,H}(I,z) = \sum_{h \in I} \pi_{-i}^{\sigma^t}(h)\left[v_i^{t,L}(\sigma^t, hz) - v_i^{t,H}(\sigma^t, h)\right]$$

$$R_i^{T,L}(a|I,z) = \frac{1}{T}\sum_{t=1}^T r_i^{t,L}(Iz,a), \; r_i^{t,L}(Iz,a) = \sum_{h \in I} \pi_{-i}^{\sigma^t}(h)\left[v_i^{t,H}(\sigma^t, hza) - v_i^{t,L}(\sigma^t, hz)\right]$$

$$v_i^{t,H}(\sigma^t, h) = \sum_{z \in Z(h)} \sigma_{P(h)}^{t,H}(z|h)v_i^{t,L}(\sigma^t, hz), \; v_i^{t,L}(\sigma^t, hz) = \sum_{a \in A(h)} \sigma_{P(h)}^{t,L}(a|h,z)v_i^{t,H}(\sigma^t, hza) \tag{25}$$

The equivalence between Equation (25) and (6) is proved in Appendix F.

Next, we propose to collect trajectories $h' \in H_{TS}$ with the sample strategy $q^t$ at each iteration $t$, and compute the corresponding **sampled** immediate counterfactual regrets $\hat{r}_i^t$ and values $\hat{v}_i^t$ as follows:

$$\hat{r}_i^{t,H}(I,z|h') = \sum_{h \in I} \frac{\pi_{-i}^{\sigma^t}(h)}{\pi^{q^t}(h)}\left[\hat{v}_i^{t,H}(\sigma^t, h, z|h') - \hat{v}_i^{t,H}(\sigma^t, h|h')\right]$$

$$\hat{r}_i^{t,L}(Iz,a|h') = \sum_{h \in I} \frac{\pi_{-i}^{\sigma^t}(h)}{\pi^{q^t}(hz)}\left[\hat{v}_i^{t,L}(\sigma^t, hz, a|h') - \hat{v}_i^{t,L}(\sigma^t, hz|h')\right] \tag{26}$$

Here, inspired by Davis et al. (2020), $\hat{v}_i^{t,H}(\sigma^t, h, z|h')$ and $\hat{v}_i^{t,L}(\sigma^t, hz, a|h')$ are incorporated with the baseline function $b_i^t$ for variance reduction: $(\hat{v}_i^{t,H}(\sigma^t, h'|h') = u_i(h'))$

$$\hat{v}_i^{t,H}(\sigma^t, h, z|h') = \frac{\delta(hz \sqsubseteq h')}{q^t(z|h)}\left[\hat{v}_i^{t,L}(\sigma^t, hz|h') - b_i^t(h,z)\right] + b_i^t(h,z)$$

$$\hat{v}_i^{t,L}(\sigma^t, hz, a|h') = \frac{\delta(hza \sqsubseteq h')}{q^t(a|h,z)}\left[\hat{v}_i^{t,H}(\sigma^t, hza|h') - b_i^t(h,z,a)\right] + b_i^t(h,z,a) \tag{27}$$

where $\delta(\cdot)$ is the indicator function. Accordingly, $\hat{v}_i^{t,H}(\sigma^t, h|h') = \sum_{z \in Z(h)} \sigma_{P(h)}^{t,H}(z|h)\hat{v}_i^{t,H}(\sigma^t, h, z|h')$ and $\hat{v}_i^{t,L}(\sigma^t, hz|h') = \sum_{a \in A(h)} \sigma_{P(h)}^{t,L}(a|h,z)\hat{v}_i^{t,L}(\sigma^t, hz, a|h')$. For superscripts on $\hat{r}$ and $\hat{v}$, we use $H$ when the agent is in state $h$ or $hza$, corresponding to high-level option choices, and $L$ when the agent is in state $hz$, corresponding to low-level action decisions.

Regarding estimators proposed in Eqs (26) and (27), we have the following theorems:

**Theorem 4.** *For all $i \in N$, $I \in \mathcal{I}_i$, $z \in Z(I)$, $a \in A(I)$, we have:*

$$\mathbb{E}_{h' \sim \pi^{q^t}(\cdot)}\left[\hat{r}_i^{t,H}(I, z|h')\right] = r_i^{t,H}(I, z), \; \mathbb{E}_{h' \sim \pi^{q^t}(\cdot)}\left[\hat{r}_i^{t,L}(Iz, a|h')\right] = r_i^{t,L}(Iz, a) \quad (28)$$

Therefore, we can acquire unbiased estimations of $R_i^T$ by substituting $r_i^t$ with $\hat{r}_i^t$ in Equation (25). This theorem is proved in Appendix G. Notably, Theorem 4 doesn't prescribe any specific form for the baseline function $b_i^t$. Yet, the baseline design can affect the sample variance of these unbiased estimators. As posited in Gibson et al. (2012), given a fixed $\epsilon > 0$, estimators with reduced variance necessitate fewer iterations to converge to an $\epsilon$-Nash equilibrium. Hence, we propose the following ideal criteria for the baseline function to minimize the sample variance:

**Theorem 5.** *If $b_i^t(h, z, a) = v_i^{t,H}(\sigma^t, hza)$ and $b_i^t(h, z) = v_i^{t,L}(\sigma^t, hz)$, for all $h \in H \backslash H_{TS}$, $z \in Z(h)$, $a \in A(h)$, we have:*

$$\mathrm{Var}_{h'}\left[\hat{v}_i^{t,H}(\sigma^t, h, z|h')|h' \sqsupseteq h\right] = \mathrm{Var}_{h'}\left[\hat{v}_i^{t,L}(\sigma^t, hz, a|h')|h' \sqsupseteq hz\right] = 0 \quad (29)$$

*Consequently, $\mathrm{Var}_{h' \sim \pi^{q^t}(\cdot)}\left[\hat{r}_i^{t,H}(I, z|h')\right]$ and $\mathrm{Var}_{h' \sim \pi^{q^t}(\cdot)}\left[\hat{r}_i^{t,L}(Iz, a|h')\right]$ are minimized with respect to $b_i^t$ for all $I \in \mathcal{I}_i$, $z \in Z(I)$, $a \in A(I)$.*

The proof can be found in Appendix H. **The ideal criteria for the baseline function proposed in Theorem 5 is incorporated into our objective design in Section 3.3.**

To sum up, by utilizing the immediate counterfactual regret estimators defined in Equations (26) and (27) and selecting baseline functions that meet the ideal criteria, we can enhance the adaptability and learning efficiency of our method (i.e., HCFR proposed in Section 3.2) through a low-variance outcome Monte Carlo sampling extension.

# F  PROOF OF EQUIVALENCE BETWEEN EQUATIONS (25) AND (6)

Through induction on the height of $h$ on the game tree, one can easily prove that:

$$v_i^{t,H}(\sigma^t, h) = \sum_{h' \in H_{TS}} \pi^{\sigma^t}(h, h')u_i(h'), \; v_i^{t,L}(\sigma^t, hz) = \sum_{h' \in H_{TS}} \pi^{\sigma^t}(hz, h')u_i(h') \quad (30)$$

Thus, we have:

$$r_i^{t,H}(I, z) = \sum_{h \in I} \pi_{-i}^{\sigma^t}(h) \sum_{h' \in H_{TS}} \pi^{\sigma^t}(hz, h')u_i(h') - \sum_{h \in I} \pi_{-i}^{\sigma^t}(h) \sum_{h' \in H_{TS}} \pi^{\sigma^t}(h, h')u_i(h')$$

$$= \pi_{-i}^{\sigma^t}(I)(u_i(\sigma^t|_{I \to z}, I) - u_i(\sigma^t, I))$$

$$r_i^{t,L}(Iz, a) = \sum_{h \in I} \pi_{-i}^{\sigma^t}(h) \sum_{h' \in H_{TS}} \pi^{\sigma^t}(hza, h')u_i(h') - \sum_{h \in I} \pi_{-i}^{\sigma^t}(h) \sum_{h' \in H_{TS}} \pi^{\sigma^t}(hz, h')u_i(h')$$

$$= \pi_{-i}^{\sigma^t}(I)(u_i(\sigma^t|_{Iz \to a}, Iz) - u_i(\sigma^t, Iz))$$

$$(31)$$

The equation above connects the definitions of $R_i^{T,H}$ and $R_i^{T,L}$ in Equation (25) and (6).

# G  PROOF OF THEOREM 4

**Lemma 4.** *For all $h \in H \backslash H_{TS}$, $z \in Z(h)$, $a \in A(h)$:*

$$E_{h'}\left[\hat{v}_i^{t,H}(\sigma^t, h, z|h')|h' \sqsupseteq h\right] = E_{h'}\left[\hat{v}_i^{t,L}(\sigma^t, hz|h')|h' \sqsupseteq hz\right]$$

$$E_{h'}\left[\hat{v}_i^{t,L}(\sigma^t, hz, a|h')|h' \sqsupseteq hz\right] = E_{h'}\left[\hat{v}_i^{t,H}(\sigma^t, hza|h')|h' \sqsupseteq hza\right]$$

$$(32)$$

*Proof.*

$$E_{h'}\left[\hat{v}_i^{t,H}(\sigma^t, h, z|h')|h' \sqsupseteq h\right]$$

$$= E_{h'}\left[\frac{\delta(hz \sqsubseteq h')}{q^t(z|h)}\left[\hat{v}_i^{t,L}(\sigma^t, hz|h') - b_i^t(h,z)\right] + b_i^t(h,z)|h' \sqsupseteq h\right]$$

$$= P(hz \sqsubseteq h'|h' \sqsupseteq h)E_{h'}\left[\frac{1}{q^t(z|h)}\left[\hat{v}_i^{t,L}(\sigma^t, hz|h') - b_i^t(h,z)\right] + b_i^t(h,z)|h' \sqsupseteq hz\right] +$$

$$\quad P(hz \not\sqsubseteq h'|h' \sqsupseteq h)b_i^t(h,z)$$

$$= q^t(z|h)\left[\frac{1}{q^t(z|h)}\left[E_{h'}(\hat{v}_i^{t,L}(\sigma^t, hz|h')|h' \sqsupseteq hz) - b_i^t(h,z)\right] + b_i^t(h,z)\right] + (1 - q^t(z|h))b_i^t(h,z)$$

$$= E_{h'}\left[\hat{v}_i^{t,L}(\sigma^t, hz|h')|h' \sqsupseteq hz\right]$$

$$\tag{33}$$

Using the definition of $\hat{v}_i^{t,L}(\sigma^t, hz, a|h')$ in Equation (27) and following the same process as above, we can get the second part of the lemma. □

Now, we present the proof of the first part of Theorem 4.

$$\mathbb{E}_{h' \sim \pi^{q^t}(\cdot)}\left[\hat{r}_i^{t,H}(I, z|h')\right] = \sum_{h \in I} \frac{\pi_{-i}^{\sigma^t}(h)}{\pi^{q^t}(h)}\left[\mathbb{E}_{h'}\left[\hat{v}_i^{t,H}(\sigma^t, h, z|h')\right] - \mathbb{E}_{h'}\left[\hat{v}_i^{t,H}(\sigma^t, h|h')\right]\right]$$

$$= \sum_{h \in I} \pi_{-i}^{\sigma^t}(h)\left[\mathbb{E}_{h'}\left[\hat{v}_i^{t,H}(\sigma^t, h, z|h')|h' \sqsupseteq h\right] - \mathbb{E}_{h'}\left[\hat{v}_i^{t,H}(\sigma^t, h|h')|h' \sqsupseteq h\right]\right]$$

$$\tag{34}$$

For the second equality, we use the following fact:

$$\mathbb{E}_{h'}\left[\hat{v}_i^{t,H}(\sigma^t, h|h')\right] = P(h' \sqsupseteq h)\mathbb{E}_{h'}\left[\hat{v}_i^{t,H}(\sigma^t, h|h')|h' \sqsupseteq h\right] + P(h' \not\sqsupseteq h)\mathbb{E}_{h'}\left[\hat{v}_i^{t,H}(\sigma^t, h|h')|h' \not\sqsupseteq h\right]$$

$$= \pi^{q^t}(h)\mathbb{E}_{h'}\left[\hat{v}_i^{t,H}(\sigma^t, h|h')|h' \sqsupseteq h\right]$$

$$\tag{35}$$

Based on Equation (27), $\mathbb{E}_{h'}\left[\hat{v}_i^{t,H}(\sigma^t, h|h')|h' \not\sqsupseteq h\right] = \mathbb{E}_{h'}\left[\hat{v}_i^{t,H}(\sigma^t, h, z|h')|h' \not\sqsupseteq h\right] = 0$. Similar with Equation (35), we can get $\mathbb{E}_{h'}\left[\hat{v}_i^{t,H}(\sigma^t, h, z|h')\right] = \pi^{q^t}(h)\mathbb{E}_{h'}\left[\hat{v}_i^{t,H}(\sigma^t, h, z|h')|h' \sqsupseteq h\right]$, which completes the proof of Equation (34).

Equation (25) and (34) show that, to prove $\mathbb{E}_{h' \sim \pi^{q^t}(\cdot)}\left[\hat{r}_i^{t,H}(I, z|h')\right] = r_i^{t,H}(I, z)$, we only need to show the following lemma:

**Lemma 5.** *For all $h \in H$, $z \in Z(h)$:*

$$E_{h'}\left[\hat{v}_i^{t,H}(\sigma^t, h, z|h')|h' \sqsupseteq h\right] = v_i^{t,L}(\sigma^t, hz), \quad E_{h'}\left[\hat{v}_i^{t,H}(\sigma^t, h|h')|h' \sqsupseteq h\right] = v_i^{t,H}(\sigma^t, h) \quad (36)$$

*Proof.* We prove this lemma by induction on the height of $h$ on the game tree. For the base case, if $(hza) \in H_{TS}$, we have:

$$E_{h'}\left[\hat{v}_i^{t,H}(\sigma^t, h, z|h')|h' \sqsupseteq h\right] = E_{h'}\left[\hat{v}_i^{t,L}(\sigma^t, hz|h')|h' \sqsupseteq hz\right]$$

$$= \sum_{a \in A(h)} \sigma_{P(h)}^{t,L}(a|h,z)E_{h'}\left[\hat{v}_i^{t,L}(\sigma^t, hz, a|h')|h' \sqsupseteq hz\right]$$

$$= \sum_{a \in A(h)} \sigma_{P(h)}^{t,L}(a|h,z)E_{h'}\left[\hat{v}_i^{t,H}(\sigma^t, hza|h')|h' \sqsupseteq hza\right]$$

$$= \sum_{a \in A(h)} \sigma_{P(h)}^{t,L}(a|h,z)u_i(hza) = v_i^{t,L}(\sigma^t, hz)$$

$$\tag{37}$$

Here, the first and third equality are due to Lemma 4, and the others are based on the corresponding definitions. Still, for this base case, we have:

$$
\begin{aligned}
E_{h'}\left[\hat{v}_i^{t,H}(\sigma^t, h|h')|h' \sqsupseteq h\right] &= \sum_{z \in Z(h)} \sigma_{P(h)}^{t,H}(z|h)E_{h'}\left[\hat{v}_i^{t,H}(\sigma^t, h, z|h')|h' \sqsupseteq h\right] \\
&= \sum_{z \in Z(h)} \sigma_{P(h)}^{t,H}(z|h)v_i^{t,L}(\sigma^t, hz) = v_i^{t,H}(\sigma^t, h)
\end{aligned}
\tag{38}
$$

where the second equality comes for Equation (37). Then, we can move on to the general case, with the hypothesis that Lemma 5 holds for the nodes lower than $h$ on the game tree:

$$
\begin{aligned}
E_{h'}\left[\hat{v}_i^{t,H}(\sigma^t, h, z|h')|h' \sqsupseteq h\right] &= E_{h'}\left[\hat{v}_i^{t,L}(\sigma^t, hz|h')|h' \sqsupseteq hz\right] \\
&= \sum_{a \in A(h)} \sigma_{P(h)}^{t,L}(a|h,z)E_{h'}\left[\hat{v}_i^{t,L}(\sigma^t, hz, a|h')|h' \sqsupseteq hz\right] \\
&= \sum_{a \in A(h)} \sigma_{P(h)}^{t,L}(a|h,z)E_{h'}\left[\hat{v}_i^{t,H}(\sigma^t, hza|h')|h' \sqsupseteq hza\right] \\
&= \sum_{a \in A(h)} \sigma_{P(h)}^{t,L}(a|h,z)v_i^{t,H}(\sigma^t, hza) = v_i^{t,L}(\sigma^t, hz)
\end{aligned}
\tag{39}
$$

where the induction hypothesis is adopted for the fourth equality. Equation (39) and (38) imply that $E_{h'}\left[\hat{v}_i^{t,H}(\sigma^t, h|h')|h' \sqsupseteq h\right] = v_i^{t,H}(\sigma^t, h)$ holds for the general case. $\qquad\square$

So far, we have proved the first part of Theorem 4, i.e., $\mathbb{E}_{h' \sim \pi^{q^t}(\cdot)}\left[\hat{r}_i^{t,H}(I, z|h')\right] = r_i^{t,H}(I, z)$. The second part, $\mathbb{E}_{h' \sim \pi^{q^t}(\cdot)}\left[\hat{r}_i^{t,L}(Iz, a|h')\right] = r_i^{t,L}(Iz, a)$, can be proved with the same process as above based on Lemma 4, so we skip the complete proof and only present the following lemma within it.

**Lemma 6.** *For all $h \in H \backslash H_{TS}$, $z \in Z(h), a \in A(h)$:*

$$
E_{h'}\left[\hat{v}_i^{t,L}(\sigma^t, hz, a|h')|h' \sqsupseteq hz\right] = v_i^{t,H}(\sigma^t, hza), \ E_{h'}\left[\hat{v}_i^{t,L}(\sigma^t, hz|h')|h' \sqsupseteq hz\right] = v_i^{t,L}(\sigma^t, hz)
\tag{40}
$$

# H   PROOF OF THEOREM 5

*Part I:*

First, we can apply the law of total variance to $\mathrm{Var}_{h' \sim \pi^{q^t}(\cdot)}\left[\hat{r}_i^{t,H}(I, z|h')\right]$, conditioning on $\delta(h' \sqsupseteq I)$ (i.e., if $h'$ is reachable from $I$), and get:

$$
\mathrm{Var}_{h' \sim \pi^{q^t}(\cdot)}\left[\hat{r}_i^{t,H}(I, z|h')\right] = \mathbb{E}\left[\mathrm{Var}_{h'}\left[\hat{r}_i^{t,H}(I, z|h')|\delta(h' \sqsupseteq I)\right]\right] + \mathrm{Var}\left[\mathbb{E}_{h'}\left[\hat{r}_i^{t,H}(I, z|h')|\delta(h' \sqsupseteq I)\right]\right]
\tag{41}
$$

The first term can be expanded as follows, where the second equality is due to $\hat{r}_i^{t,H}(I, z|h') = 0$ when $h' \not\sqsupseteq I$.

$$
\begin{aligned}
&\mathbb{E}\left[\mathrm{Var}_{h'}\left[\hat{r}_i^{t,H}(I, z|h')|\delta(h' \sqsupseteq I)\right]\right] \\
&= P(h' \sqsupseteq I)\mathrm{Var}_{h'}\left[\hat{r}_i^{t,H}(I, z|h')|h' \sqsupseteq I\right] + P(h' \not\sqsupseteq I)\mathrm{Var}_{h'}\left[\hat{r}_i^{t,H}(I, z|h')|h' \not\sqsupseteq I\right] \\
&= P(h' \sqsupseteq I)\mathrm{Var}_{h'}\left[\hat{r}_i^{t,H}(I, z|h')|h' \sqsupseteq I\right]
\end{aligned}
\tag{42}
$$

The second term can be converted as follows, based on the fact that $\mathbb{E}_{h'}(\hat{r}_i^{t,H}(I, z|h')|\delta(h' \sqsupseteq I)) = \frac{r_i^{t,H}(I,z)}{P(h' \sqsupseteq I)}$ (i.e., $\mathbb{E}_{h'}(\hat{r}_i^{t,H}(I, z|h')|h' \sqsupseteq I))$ with probability $P(h' \sqsupseteq I)$, and $\mathbb{E}_{h'}(\hat{r}_i^{t,H}(I, z|h')|$

$\delta(h' \sqsupseteq I)) = 0$ (i.e., $\mathbb{E}_{h'}(\hat{r}_i^{t,H}(I, z|h')|h' \not\sqsupseteq I))$ with probability $1 - P(h' \sqsupseteq I)$.

$$\mathrm{Var}\left[\mathbb{E}_{h'}\left[\hat{r}_i^{t,H}(I, z|h')|\delta(h' \sqsupseteq I)\right]\right]$$

$$= \mathbb{E}\left[\left[\mathbb{E}_{h'}(\hat{r}_i^{t,H}(I, z|h')|\delta(h' \sqsupseteq I))\right]^2\right] - \left[\mathbb{E}\left[\mathbb{E}_{h'}(\hat{r}_i^{t,H}(I, z|h')|\delta(h' \sqsupseteq I))\right]\right]^2 \qquad (43)$$

$$= \frac{1 - P(h' \sqsupseteq I)}{P(h' \sqsupseteq I)}(r_i^{t,H}(I, z))^2$$

Note that $\frac{1-P(h'\sqsupseteq I)}{P(h'\sqsupseteq I)}(r_i^{t,H}(I, z))^2$ and $P(h' \sqsupseteq I)$ is not affected by $b_i^t$, so we focus on $\mathrm{Var}_{h'}\left[\hat{r}_i^{t,H}(I, z|h')|h' \sqsupseteq I\right]$ in Equation (42). Applying the law of total variance:

$$\mathrm{Var}_{h'}\left[\hat{r}_i^{t,H}(I, z|h')|h' \sqsupseteq I\right]$$

$$= \mathbb{E}_{h\in I}\left[\mathrm{Var}_{h'}\left[\hat{r}_i^{t,H}(I, z|h')|h' \sqsupseteq h\right]\right] + \mathrm{Var}_{h\in I}\left[\mathbb{E}_{h'}\left[\hat{r}_i^{t,H}(I, z|h')|h' \sqsupseteq h\right]\right] \qquad (44)$$

$$\geq \mathrm{Var}_{h\in I}\left[\mathbb{E}_{h'}\left[\hat{r}_i^{t,H}(I, z|h')|h' \sqsupseteq h\right]\right]$$

Fix $h \in I$, $\mathbb{E}_{h'}\left[\hat{r}_i^{t,H}(I, z|h')|h' \sqsupseteq h\right] = \frac{\pi_{-i}^{\sigma^t}(h)}{\pi^{q^t}(h)}\left[v_i^{t,L}(\sigma^t, hz) - v_i^{t,H}(\sigma^t, h)\right]$, based on the definition of $\hat{r}_i^{t,H}(I, z|h')$ and Lemma 5. Thus, the second term in Equation (44) is irrelevant to $b_i^t$. According to Equation (41)-(44), we conclude that the minimum of $\mathrm{Var}_{h'\sim\pi^{q^t}(\cdot)}\left[\hat{r}_i^{t,H}(I, z|h')\right]$ with respect to $b_i^t$ can be achieved when $\mathbb{E}_{h\in I}\left[\mathrm{Var}_{h'}\left[\hat{r}_i^{t,H}(I, z|h')|h' \sqsupseteq h\right]\right] = 0$. Following the same process, we can show that the minimum of $\mathrm{Var}_{h'\sim\pi^{q^t}(\cdot)}\left[\hat{r}_i^{t,L}(Iz, a|h')\right]$ with respect to $b_i^t$ can be achieved when $\mathbb{E}_{h\in I}\left[\mathrm{Var}_{h'}\left[\hat{r}_i^{t,L}(Iz, a|h')|h' \sqsupseteq hz\right]\right] = 0$.

**Lemma 7.** *If* $\mathrm{Var}_{h'}\left[\hat{v}_i^{t,H}(\sigma^t, h, z|h')|h' \sqsupseteq h\right] = \mathrm{Var}_{h'}\left[\hat{v}_i^{t,L}(\sigma^t, hz, a|h')|h' \sqsupseteq hz\right] = 0$, *for all* $h \in H\backslash H_{TS}$, $z \in Z(h)$, $a \in A(h)$, *then* $\mathbb{E}_{h\in I}\left[\mathrm{Var}_{h'}\left[\hat{r}_i^{t,H}(I, z|h')|h' \sqsupseteq h\right]\right] = \mathbb{E}_{h\in I}\left[\mathrm{Var}_{h'}\left[\hat{r}_i^{t,L}(Iz, a|h')|h' \sqsupseteq hz\right]\right] = 0, \forall I \in \mathcal{I}_i, z \in Z(I), a \in A(I)$.

*Proof.* Pick any $h \in H\backslash H_{TS}$, $z \in Z(h)$. Based on Lemma 5, $E_{h'}\left[\hat{v}_i^{t,H}(\sigma^t, h, z|h')|h' \sqsupseteq h\right] = v_i^{t,L}(\sigma^t, hz)$. If $\mathrm{Var}_{h'}\left[\hat{v}_i^{t,H}(\sigma^t, h, z|h')|h' \sqsupseteq h\right] = 0$, then $\hat{v}_i^{t,H}(\sigma^t, h, z|h') = v_i^{t,L}(\sigma^t, hz)$, $\forall h' \sqsupseteq h$. It follows that $\hat{v}_i^{t,H}(\sigma^t, h|h') = v_i^{t,H}(\sigma^t, h)$, $\forall h' \sqsupseteq h$, based on the definitions of $v_i^{t,H}(\sigma^t, h)$ and $\hat{v}_i^{t,H}(\sigma^t, h|h')$. Now, for any $I \in \mathcal{I}_i$, $h \in I$, $h' \sqsupseteq h$:

$$\hat{r}_i^{t,H}(I, z|h') = \sum_{h''\in I} \frac{\pi_{-i}^{\sigma^t}(h'')}{\pi^{q^t}(h'')}\left[\hat{v}_i^{t,H}(\sigma^t, h'', z|h') - \hat{v}_i^{t,H}(\sigma^t, h''|h')\right]$$

$$= \frac{\pi_{-i}^{\sigma^t}(h)}{\pi^{q^t}(h)}\left[\hat{v}_i^{t,H}(\sigma^t, h, z|h') - \hat{v}_i^{t,H}(\sigma^t, h|h')\right] \qquad (45)$$

$$= \frac{\pi_{-i}^{\sigma^t}(h)}{\pi^{q^t}(h)}\left[v_i^{t,L}(\sigma^t, hz) - v_i^{t,H}(\sigma^t, h)\right]$$

Thus, $\mathrm{Var}_{h'}\left[\hat{r}_i^{t,H}(I, z|h')|h' \sqsupseteq h\right] = 0$, $\forall I \in \mathcal{I}_i$, $h \in I$. Then, it follows that for any $I$, $\mathbb{E}_{h\in I}\left[\mathrm{Var}_{h'}\left[\hat{r}_i^{t,H}(I, z|h')|h' \sqsupseteq h\right]\right] = 0$. With the same process as above, we can show the second part of Lemma 7. $\square$

Given the discussions above, to complete the proof of Theorem 5, we need to further show that, $\forall i \in N$, if $b_i^t(h, z, a) = v_i^{t,H}(\sigma^t, hza)$ and $b_i^t(h, z) = v_i^{t,L}(\sigma^t, hz)$, for all $h \in H\backslash H_{TS}$, $z \in Z(h)$, $a \in$

$A(h)$, we have $\mathrm{Var}_{h'}\left[\hat{v}_i^{t,H}(\sigma^t, h, z|h')|h' \sqsupseteq h\right] = \mathrm{Var}_{h'}\left[\hat{v}_i^{t,L}(\sigma^t, hz, a|h')|h' \sqsupseteq hz\right] = 0$, for all $h \in H\backslash H_{TS}$, $z \in Z(h)$, $a \in A(h)$.

**Part II:**

**Lemma 8.** *For any $i \in N$, $h \in H\backslash H_{TS}$, $z \in Z(h), a \in A(h)$ and any baseline function $b_i^t$:*

$$
\begin{aligned}
\mathrm{Var}_{h'}\left[\hat{v}_i^{t,H}(\sigma^t, h|h')|h' \sqsupseteq h\right] &= \sum_{z \in Z(h)} \frac{(\sigma_{P(h)}^{t,H}(z|h))^2}{q^t(z|h)}\mathrm{Var}_{h'}\left[\hat{v}_i^{t,L}(\sigma^t, hz|h')|h' \sqsupseteq hz\right] \\
&\quad + \mathrm{Var}_{z \sim q^t(\cdot|h)}\left[\frac{\sigma_{P(h)}^{t,H}(z|h)}{q^t(z|h)}(v_i^{t,L}(\sigma^t, hz) - b_i^t(h,z))\right] \\
\mathrm{Var}_{h'}\left[\hat{v}_i^{t,L}(\sigma^t, hz|h')|h' \sqsupseteq hz\right] &= \sum_{a \in A(h)} \frac{(\sigma_{P(h)}^{t,L}(a|h,z))^2}{q^t(a|h,z)}\mathrm{Var}_{h'}\left[\hat{v}_i^{t,H}(\sigma^t, hza|h')|h' \sqsupseteq hza\right] \\
&\quad + \mathrm{Var}_a\left[\frac{\sigma_{P(h)}^{t,L}(a|h,z)}{q^t(a|h,z)}(v_i^{t,H}(\sigma^t, hza) - b_i^t(h,z,a))\right]
\end{aligned}
\tag{46}
$$

*Proof.* By conditioning on the option choice at $h$, we apply the law of total variance to $\mathrm{Var}_{h'}\left[\hat{v}_i^{t,H}(\sigma^t, h|h')|h' \sqsupseteq h\right]$:

$$
\begin{aligned}
\mathrm{Var}_{h'}\left[\hat{v}_i^{t,H}(\sigma^t, h|h')|h' \sqsupseteq h\right] &= \mathbb{E}_{z \sim q^t(\cdot|h)}\left[\mathrm{Var}_{h'}\left[\hat{v}_i^{t,H}(\sigma^t, h|h')|h' \sqsupseteq hz\right]\right] + \\
&\quad \mathrm{Var}_{z \sim q^t(\cdot|h)}\left[\mathbb{E}_{h'}\left[\hat{v}_i^{t,H}(\sigma^t, h|h')|h' \sqsupseteq hz\right]\right]
\end{aligned}
\tag{47}
$$

According to the definition of $\hat{v}_i^{t,H}(\sigma^t, h|h')$ and the fact that $h' \sqsupseteq hz$, we have:

$$
\begin{aligned}
&\mathrm{Var}_{h'}\left[\hat{v}_i^{t,H}(\sigma^t, h|h')|h' \sqsupseteq hz\right] \\
&= \mathrm{Var}_{h'}\left[\frac{\sigma_{P(h)}^{t,H}(z|h)}{q^t(z|h)}\left[\hat{v}_i^{t,L}(\sigma^t, hz|h') - b_i^t(h,z)\right] + \sum_{z' \in Z(h)} \sigma_{P(h)}^{t,H}(z'|h)b_i^t(h,z')|h' \sqsupseteq hz\right] \\
&= \left[\frac{\sigma_{P(h)}^{t,H}(z|h)}{q^t(z|h)}\right]^2 \mathrm{Var}_{h'}\left[\hat{v}_i^{t,L}(\sigma^t, hz|h')|h' \sqsupseteq hz\right]
\end{aligned}
$$

$$
\mathbb{E}_{z \sim q^t(\cdot|h)}\left[\mathrm{Var}_{h'}\left[\hat{v}_i^{t,H}(\sigma^t, h|h')|h' \sqsupseteq hz\right]\right] = \sum_{z \in Z(h)} \frac{(\sigma_{P(h)}^{t,H}(z|h))^2}{q^t(z|h)}\mathrm{Var}_{h'}\left[\hat{v}_i^{t,L}(\sigma^t, hz|h')|h' \sqsupseteq hz\right]
\tag{48}
$$

Then, we analyze the second term in Equation (47):

$$
\begin{aligned}
&\mathbb{E}_{h'}\left[\hat{v}_i^{t,H}(\sigma^t, h|h')|h' \sqsupseteq hz\right] \\
&= \frac{\sigma_{P(h)}^{t,H}(z|h)}{q^t(z|h)}\left[\mathbb{E}_{h'}\left[\hat{v}_i^{t,L}(\sigma^t, hz|h')|h' \sqsupseteq hz\right] - b_i^t(h,z)\right] + \sum_{z' \in Z(h)} \sigma_{P(h)}^{t,H}(z'|h)b_i^t(h,z') \\
&= \frac{\sigma_{P(h)}^{t,H}(z|h)}{q^t(z|h)}\left[v_i^{t,L}(\sigma^t, hz) - b_i^t(h,z)\right] + \sum_{z' \in Z(h)} \sigma_{P(h)}^{t,H}(z'|h)b_i^t(h,z')
\end{aligned}
$$

$$
\mathrm{Var}_{z \sim q^t(\cdot|h)}\left[\mathbb{E}_{h'}\left[\hat{v}_i^{t,H}(\sigma^t, h|h')|h' \sqsupseteq hz\right]\right] = \mathrm{Var}_{z \sim q^t(\cdot|h)}\left[\frac{\sigma_{P(h)}^{t,H}(z|h)}{q^t(z|h)}\left[v_i^{t,L}(\sigma^t, hz) - b_i^t(h,z)\right]\right]
\tag{49}
$$

Based on Equation (47)-(49), we can get the first part of Lemma 8. The second part can be obtained similarly. $\qquad\square$

Lemma 8 illustrates the outcome of a single-step lookahead from state $h$. Employing this in an inductive manner, we can derive the complete expansion of $\text{Var}_{h'}\left[\hat{v}_i^{t,H}(\sigma^t, h|h')|h' \sqsupseteq h\right]$ on the game tree as the following lemma:

**Lemma 9.** *For any $i \in N$, $h \in H$ and any baseline function $b_i^t$:*

$$\text{Var}_{h'}\left[\hat{v}_i^{t,H}(\sigma^t, h|h')|h' \sqsupseteq h\right] = \sum_{h'' \sqsupseteq h} \frac{(\pi^{\sigma^t}(h, h''))^2}{\pi^{q^t}(h, h'')} f(h'')$$

$$f(h'') = \text{Var}_z\left[\frac{\sigma_{P(h'')}^{t,H}(z|h'')}{q^t(z|h'')}(v_i^{t,L}(\sigma^t, h''z) - b_i^t(h'', z))\right] + \quad (50)$$

$$\sum_{z \in Z(h'')} \frac{(\sigma_{P(h'')}^{t,H}(z|h''))^2}{q^t(z|h'')}\text{Var}_a\left[\frac{\sigma_{P(h'')}^{t,L}(a|h'', z)}{q^t(a|h'', z)}(v_i^{t,H}(\sigma^t, h''za) - b_i^t(h'', z, a))\right]$$

*Proof.* We proof this lemma through an induction on the height of $h$ on the game tree. For the base case, $h \in H_{TS}$, then $Z(h) = A(h) = \emptyset$, so $f(h'') = 0$, $\forall h'' \sqsupseteq h$. In addition, we have $\text{Var}_{h'}\left[\hat{v}_i^{t,H}(\sigma^t, h|h')|h' \sqsupseteq h\right] = \text{Var}_{h'}\left[\hat{v}_i^{t,H}(\sigma^t, h|h')|h' = h\right] = 0$. Thus, the lemma holds for the base case.

For the general case, $h \in H \backslash H_{TS}$, we apply Lemma 8 and get:

$$\text{Var}_{h'}\left[\hat{v}_i^{t,H}(\sigma^t, h|h')|h' \sqsupseteq h\right] = \text{Var}_{z \sim q^t(\cdot|h)}\left[\frac{\sigma_{P(h)}^{t,H}(z|h)}{q^t(z|h)}(v_i^{t,L}(\sigma^t, hz) - b_i^t(h, z))\right]$$

$$+ \sum_{z \in Z(h)} \frac{(\sigma_{P(h)}^{t,H}(z|h))^2}{q^t(z|h)}\text{Var}_{a \sim q^t(\cdot|h,z)}\left[\frac{\sigma_{P(h)}^{t,L}(a|h, z)}{q^t(a|h, z)}(v_i^{t,H}(\sigma^t, hza) - b_i^t(h, z, a))\right]$$

$$+ \sum_{(z,a) \in \widetilde{A}(h)} \frac{(\sigma_{P(h)}^t((z,a)|h))^2}{q^t((z,a)|h)}\text{Var}_{h'}\left[\hat{v}_i^{t,H}(\sigma^t, hza|h')|h' \sqsupseteq hza\right] \quad (51)$$

$$= f(h) + \sum_{(z,a) \in \widetilde{A}(h)} \frac{(\sigma_{P(h)}^t((z,a)|h))^2}{q^t((z,a)|h)}\text{Var}_{h'}\left[\hat{v}_i^{t,H}(\sigma^t, hza|h')|h' \sqsupseteq hza\right]$$

where the first equality is the result of the sequential use of the two formulas in Lemma 8 and the second equality is based on the definition of $f(h)$. Next, we apply the induction hypothesis on $hza$, i.e., a node lower than $h$ on the game tree, and get:

$$\text{Var}_{h'}\left[\hat{v}_i^{t,H}(\sigma^t, hza|h')|h' \sqsupseteq hza\right] = \sum_{h'' \sqsupseteq hza} \frac{(\pi^{\sigma^t}(hza, h''))^2}{\pi^{q^t}(hza, h'')} f(h'') \quad (52)$$

By integrating Equation (51) and (52), we can get:

$$\text{Var}_{h'}\left[\hat{v}_i^{t,H}(\sigma^t, h|h')|h' \sqsupseteq h\right] = f(h) + \sum_{(z,a)} \frac{(\sigma_{P(h)}^t((z,a)|h))^2}{q^t((z,a)|h)} \sum_{h'' \sqsupseteq hza} \frac{(\pi^{\sigma^t}(hza, h''))^2}{\pi^{q^t}(hza, h'')} f(h'')$$

$$= f(h) + \sum_{h'' \sqsupset h} \frac{(\pi^{\sigma^t}(h, h''))^2}{\pi^{q^t}(h, h'')} f(h'')$$

$$= \sum_{h'' \sqsupseteq h} \frac{(\pi^{\sigma^t}(h, h''))^2}{\pi^{q^t}(h, h'')} f(h'')$$

$$(53)$$

For the second equality, we use the definitions of $\pi^{\sigma^t}(h, h'')$ and $\pi^{q^t}(h, h'')$, and the fact that they equal 1 when $h'' = h$. $\qquad\square$

Before moving to the final proof, we introduce another lemma as follows.

**Lemma 10.** *For any $i \in N$, $h \in H \backslash H_{TS}$, $z \in Z(h)$ and any baseline function $b_i^t$:*

$$\mathrm{Var}_{h'}\left[\hat{v}_i^{t,L}(\sigma^t, hz|h')|h' \sqsupseteq hz\right] \leq \sum_{\substack{a \in A(h), \\ h'' \sqsupseteq hza, \\ z'' \in Z(h'')}} \frac{(\pi^{\sigma^t}(hz, h''z''))^2}{\pi^{q^t}(hz, h''z'')}\left[v_i^{t,L}(\sigma^t, h''z'') - b_i^t(h'', z'')\right]^2$$

$$+ \sum_{\substack{h''z'' \sqsupseteq hz, \\ a'' \in A(h'')}} \frac{(\pi^{\sigma^t}(hz, h''z''a''))^2}{\pi^{q^t}(hz, h''z''a'')}\left[v_i^{t,H}(\sigma^t, h''z''a'') - b_i^t(h'', z'', a'')\right]^2 \tag{54}$$

*Proof.* Applying the fact $\mathrm{Var}(X) = \mathbb{E}(X^2) - (\mathbb{E}(X))^2 \leq \mathbb{E}(X^2)$ to both variance terms of Equation (50) and after rearranging the terms, we arrive at the following expression:

$$\mathrm{Var}_{h'}\left[\hat{v}_i^{t,H}(\sigma^t, h|h')|h' \sqsupseteq h\right] \leq \sum_{\substack{h'' \sqsupseteq h, \\ z'' \in Z(h'')}} \frac{(\pi^{\sigma^t}(h, h''z''))^2}{\pi^{q^t}(h, h''z'')}\left[v_i^{t,L}(\sigma^t, h''z'') - b_i^t(h'', z'')\right]^2$$

$$+ \sum_{\substack{h'' \sqsupseteq h, \\ (z'',a'') \in \widetilde{A}(h'')}} \frac{(\pi^{\sigma^t}(h, h''z''a''))^2}{\pi^{q^t}(h, h''z''a'')}\left[v_i^{t,H}(\sigma^t, h''z''a'') - b_i^t(h'', z'', a'')\right]^2 \tag{55}$$

Note that the equation above holds for any $h \in H$. Then, to get an upper bound of $\mathrm{Var}_{h'}\left[\hat{v}_i^{t,L}(\sigma^t, hz|h')|h' \sqsupseteq hz\right]$, we go back to Lemma 8 and apply Equation (55) and $\mathrm{Var}(X) \leq \mathbb{E}(X^2)$ to its first and second term, respectively. After rearranging, we can get:

$$\mathrm{Var}_{h'}\left[\hat{v}_i^{t,L}(\sigma^t, hz|h')|h' \sqsupseteq hz\right] \leq \sum_{\substack{a \in A(h), \\ h'' \sqsupseteq hza, \\ z'' \in Z(h'')}} \frac{(\pi^{\sigma^t}(hz, h''z''))^2}{\pi^{q^t}(hz, h''z'')}\left[v_i^{t,L}(\sigma^t, h''z'') - b_i^t(h'', z'')\right]^2$$

$$+ \sum_{\substack{a \in A(h), \\ h'' \sqsupseteq hza, \\ (z'',a'') \in \widetilde{A}(h'')}} \frac{(\pi^{\sigma^t}(hz, h''z''a''))^2}{\pi^{q^t}(hz, h''z''a'')}\left[v_i^{t,H}(\sigma^t, h''z''a'') - b_i^t(h'', z'', a'')\right]^2$$

$$+ \sum_{a \in A(h)} \frac{(\sigma_{P(h)}^{t,L}(a|h,z))^2}{q^t(a|h,z)}\left[v_i^{t,H}(\sigma^t, hza) - b_i^t(h, z, a)\right]^2 \tag{56}$$

We note that the second term of Equation (54) can be obtained by combining the last two terms of Equation (56). The second and third term of Equation (56) correspond to the sum over $h''z'' \sqsupseteq hz$, $a'' \in A(h'')$ and $h''z'' = hz$, $a'' \in A(h'')$, respectively. $\qquad\square$

Based on the discussions above, we give out the upper bound of $\mathrm{Var}_{h'}\left[\hat{v}_i^{t,H}(\sigma^t, h, z|h')|h' \sqsupseteq h\right]$ as the following lemma:

**Lemma 11.** *For any $i \in N$, $h \in H \backslash H_{TS}$, $z \in Z(h)$ and any baseline function $b_i^t$:*

$$\mathrm{Var}_{h'}\left[\hat{v}_i^{t,H}(\sigma^t, h, z|h')|h' \sqsupseteq h\right] \leq \frac{1}{q^t(z|h)} \sum_{h''z'' \sqsupseteq hz} \frac{(\pi^{\sigma^t}(hz, h''z''))^2}{\pi^{q^t}(hz, h''z'')}\left[v_i^{t,L}(\sigma^t, h''z'') - b_i^t(h'', z'')\right]^2$$

$$+ \frac{1}{q^t(z|h)} \sum_{\substack{h''z'' \sqsupseteq hz, \\ a'' \in A(h'')}} \frac{(\pi^{\sigma^t}(hz, h''z''a''))^2}{\pi^{q^t}(hz, h''z''a'')}\left[v_i^{t,H}(\sigma^t, h''z''a'') - b_i^t(h'', z'', a'')\right]^2 \tag{57}$$

*Proof.*

$$\text{Var}_{h'}\left[\hat{v}_i^{t,H}(\sigma^t, h, z|h')|h' \sqsupseteq h\right]$$

$$= \text{Var}_{h'}\left[\frac{\delta(hz \sqsubseteq h')}{q^t(z|h)}\left[\hat{v}_i^{t,L}(\sigma^t, hz|h') - b_i^t(h, z)\right]|h' \sqsupseteq h\right]$$

$$= \mathbb{E}\left[\text{Var}_{h'}\left[\frac{\delta(hz \sqsubseteq h')}{q^t(z|h)}\left[\hat{v}_i^{t,L}(\sigma^t, hz|h') - b_i^t(h, z)\right]|h' \sqsupseteq h, \delta(hz \sqsubseteq h')\right]\right] + \qquad (58)$$

$$\text{Var}\left[\mathbb{E}_{h'}\left[\frac{\delta(hz \sqsubseteq h')}{q^t(z|h)}\left[\hat{v}_i^{t,L}(\sigma^t, hz|h') - b_i^t(h, z)\right]|h' \sqsupseteq h, \delta(hz \sqsubseteq h')\right]\right]$$

Here, we apply the definition of $\hat{v}_i^{t,H}(\sigma^t, h, z|h')$ to get the first equality, and the law of total variance conditioned on $\delta(hz \sqsubseteq h')$ (given $h \sqsubseteq h'$) to get the second equality. Next, we analyze the two terms in the third and fourth line of Equation (58) separately.

$$\mathbb{E}\left[\text{Var}_{h'}\left[\frac{\delta(hz \sqsubseteq h')}{q^t(z|h)}\left[\hat{v}_i^{t,L}(\sigma^t, hz|h') - b_i^t(h, z)\right]|h' \sqsupseteq h, \delta(hz \sqsubseteq h')\right]\right]$$

$$= P(hz \sqsubseteq h'|h \sqsubseteq h')\text{Var}_{h'}\left[\frac{1}{q^t(z|h)}\left[\hat{v}_i^{t,L}(\sigma^t, hz|h') - b_i^t(h, z)\right]|h' \sqsupseteq hz\right]$$

$$= q^t(z|h)\text{Var}_{h'}\left[\frac{1}{q^t(z|h)}\left[\hat{v}_i^{t,L}(\sigma^t, hz|h') - b_i^t(h, z)\right]|h' \sqsupseteq hz\right] \qquad (59)$$

$$= \frac{1}{q^t(z|h)}\text{Var}_{h'}\left[\hat{v}_i^{t,L}(\sigma^t, hz|h')|h' \sqsupseteq hz\right]$$

Note that $\delta(hz \sqsubseteq h')$ can be 0 or 1 (with probability $P(hz \sqsubseteq h'|h \sqsubseteq h')$), and the variance equals 0 when $\delta(hz \sqsubseteq h') = 0$, so we get the first equality in Equation (59). Similarly, we can get:

$$\text{Var}\left[\mathbb{E}_{h'}\left[\frac{\delta(hz \sqsubseteq h')}{q^t(z|h)}\left[\hat{v}_i^{t,L}(\sigma^t, hz|h') - b_i^t(h, z)\right]|h' \sqsupseteq h, \delta(hz \sqsubseteq h')\right]\right]$$

$$\leq \mathbb{E}\left[\left[\mathbb{E}_{h'}\left[\frac{\delta(hz \sqsubseteq h')}{q^t(z|h)}\left[\hat{v}_i^{t,L}(\sigma^t, hz|h') - b_i^t(h, z)\right]|h' \sqsupseteq h, \delta(hz \sqsubseteq h')\right]\right]^2\right]$$

$$= q^t(z|h)\left[\mathbb{E}_{h'}\left[\frac{1}{q^t(z|h)}\left[\hat{v}_i^{t,L}(\sigma^t, hz|h') - b_i^t(h, z)\right]|h' \sqsupseteq hz\right]\right]^2 \qquad (60)$$

$$= \frac{1}{q^t(z|h)}\left[\mathbb{E}_{h'}\left[\hat{v}_i^{t,L}(\sigma^t, hz|h')|h' \sqsupseteq hz\right] - b_i^t(h, z)\right]^2$$

$$= \frac{1}{q^t(z|h)}\left[v_i^{t,L}(\sigma^t, hz) - b_i^t(h, z)\right]^2$$

Integrating Equation (58)-(60) and utilizing the upper bound proposed in Lemma 10, we can get:

$$\text{Var}_{h'}\left[\hat{v}_i^{t,H}(\sigma^t, h, z|h')|h' \sqsupseteq h\right] \leq \frac{1}{q^t(z|h)}\left[v_i^{t,L}(\sigma^t, hz) - b_i^t(h, z)\right]^2 +$$

$$\frac{1}{q^t(z|h)}\sum_{\substack{a \in A(h), \\ h'' \sqsupseteq hza, \\ z'' \in Z(h'')}}\frac{(\pi^{\sigma^t}(hz, h''z''))^2}{\pi^{q^t}(hz, h''z'')}\left[v_i^{t,L}(\sigma^t, h''z'') - b_i^t(h'', z'')\right]^2 + \qquad (61)$$

$$\frac{1}{q^t(z|h)}\sum_{\substack{h''z'' \sqsupseteq hz, \\ a'' \in A(h'')}}\frac{(\pi^{\sigma^t}(hz, h''z''a''))^2}{\pi^{q^t}(hz, h''z''a'')}\left[v_i^{t,H}(\sigma^t, h''z''a'') - b_i^t(h'', z'', a'')\right]^2$$

Note that the sum of the first two terms of Equation (61) equals the first term of Equation (57). The first and second term of Equation (61) correspond to the sum over $h''z'' = hz$ and $h''z'' \sqsupseteq hz$, respectively. $\qquad \square$

Similarly, we can derive the upper bound for $\text{Var}_{h'}\left[\hat{v}_i^{t,L}(\sigma^t, hz, a|h')|h' \sqsupseteq hz\right]$ shown as follows.

**Lemma 12.** *For any $i \in N$, $h \in H \backslash H_{TS}$, $z \in Z(h)$, $a \in A(h)$ and any baseline function $b_i^t$:*

$$\mathrm{Var}_{h'}\left[\hat{v}_i^{t,L}(\sigma^t, hz, a|h')|h' \sqsupseteq hz\right]$$

$$\leq \frac{1}{q^t(a|h,z)} \sum_{h''z''a'' \sqsupseteq hza} \frac{(\pi^{\sigma^t}(hza, h''z''a''))^2}{\pi^{q^t}(hza, h''z''a'')} \left[v_i^{t,H}(\sigma^t, h''z''a'') - b_i^t(h'', z'', a'')\right]^2 + \tag{62}$$

$$\frac{1}{q^t(a|h,z)} \sum_{\substack{h'' \sqsupseteq hza, \\ z'' \in Z(h'')}} \frac{(\pi^{\sigma^t}(hza, h''z''))^2}{\pi^{q^t}(hza, h''z'')} \left[v_i^{t,L}(\sigma^t, h''z'') - b_i^t(h'', z'')\right]^2$$

*Proof.* By applying the law of total variance conditioned on $\delta(hza \sqsubseteq h')$ (given $hz \sqsubseteq h'$) and following the same process as Equation (58)-(60), we can get:

$$\mathrm{Var}_{h'}\left[\hat{v}_i^{t,L}(\sigma^t, hz, a|h')|h' \sqsupseteq hz\right] \leq \frac{1}{q^t(a|h,z)}\left[v_i^{t,H}(\sigma^t, hza) - b_i^t(h, z, a)\right]^2 + \tag{63}$$

$$\frac{1}{q^t(a|h,z)}\mathrm{Var}_{h'}\left[\hat{v}_i^{t,H}(\sigma^t, hza|h')|h' \sqsupseteq hza\right]$$

Then, we can apply the upper bound shown as Equation (55) and get:

$$\mathrm{Var}_{h'}\left[\hat{v}_i^{t,L}(\sigma^t, hz, a|h')|h' \sqsupseteq hz\right] \leq \frac{1}{q^t(a|h,z)}\left[v_i^{t,H}(\sigma^t, hza) - b_i^t(h, z, a)\right]^2 + $$

$$\frac{1}{q^t(a|h,z)} \sum_{\substack{h'' \sqsupseteq hza, \\ (z'',a'') \in \widetilde{A}(h'')}} \frac{(\pi^{\sigma^t}(hza, h''z''a''))^2}{\pi^{q^t}(hza, h''z''a'')} \left[v_i^{t,H}(\sigma^t, h''z''a'') - b_i^t(h'', z'', a'')\right]^2 + \tag{64}$$

$$\frac{1}{q^t(a|h,z)} \sum_{\substack{h'' \sqsupseteq hza, \\ z'' \in Z(h'')}} \frac{(\pi^{\sigma^t}(hza, h''z''))^2}{\pi^{q^t}(hza, h''z'')} \left[v_i^{t,L}(\sigma^t, h''z'') - b_i^t(h'', z'')\right]^2$$

Again, we can combine the first two terms of Equation (64) and get the first term of the right hand side of Equation (62), since the first term of Equation (64) corresponds to the case that $h'' = h$, $h''z''a'' \sqsupseteq hza$ and the second term is equivalent to the sum over $h'' \sqsupseteq h$, $h''z''a'' \sqsupseteq hza$. $\qquad\square$

Finally, with Lemma 11 - 12 and the fact that variance cannot be negative, we can claim: $\forall\, i \in N$, if $b_i^t(h, z, a) = v_i^{t,H}(\sigma^t, hza)$ and $b_i^t(h, z) = v_i^{t,L}(\sigma^t, hz)$, for all $h \in H \backslash H_{TS}$, $z \in Z(h)$, $a \in A(h)$, we have $\mathrm{Var}_{h'}\left[\hat{v}_i^{t,H}(\sigma^t, h, z|h')|h' \sqsupseteq h\right] = \mathrm{Var}_{h'}\left[\hat{v}_i^{t,L}(\sigma^t, hz, a|h')|h' \sqsupseteq hz\right] = 0$, for all $h \in H \backslash H_{TS}$, $z \in Z(h)$, $a \in A(h)$. This completes the proof of Theorem 5, according to the last paragraph of Part I in this section.

## I  PROOF OF PROPOSITION 2

We start from the definition:

$$\mathcal{L}_{R,i}^{t,H} = \mathcal{L}(R_{i,\theta}^{t,H}) = \mathbb{E}_{(I, \hat{r}_i^{t',H}) \sim \tau_R^i}\left[\sum_{z \in Z(I)} (R_{i,\theta}^{t,H}(z|I) - \hat{r}_i^{t',H}(I, z))^2\right]$$

$$= \frac{1}{norm} \sum_{t'=1}^{t} \sum_{I \in \mathcal{I}_i} \sum_{k=1}^{K} x_{t'}^k(I) \left[\sum_{z \in Z(I)} (R_{i,\theta}^{t,H}(z|I) - \hat{r}_i^{t',H}(I, z))^2\right] \tag{65}$$

Here, $x_{t'}^k(I)$ denotes whether $I$ is visited in the $k$-th sampled trajectory at iteration $t'$, and $norm = \sum_{t'=1}^{t} \sum_{I \in \mathcal{I}_i} \sum_{k=1}^{K} x_{t'}^k(I)$ serves as the normalizing factor.

Let $R_{i,*}^{t,H}$ denote a minimal point of $\mathcal{L}(R_{i,\theta}^{t,H})$. Utilizing the first-order necessary condition for optimality, we obtain: $\nabla\mathcal{L}(R_{i,*}^{t,H}) = 0$. Thus, for the $(I, z)$ entry of $R_{i,*}^{t,H}$, we deduce:

$$\frac{\partial\mathcal{L}(R_{i,*}^{t,H})}{\partial R_{i,\theta}^{t,H}(z|I)} = \frac{2}{norm}\sum_{t'=1}^{t}\sum_{k=1}^{K}x_{t'}^{k}(I)(R_{i,*}^{t,H}(z|I) - \hat{r}_{i}^{t',H}(I, z)) = 0$$

$$R_{i,*}^{t,H}(z|I) = \frac{1}{norm'}\sum_{t'=1}^{t}\sum_{k=1}^{K}x_{t'}^{k}(I)\hat{r}_{i}^{t',H}(I, z) = \frac{1}{norm'}\sum_{t'=1}^{t}\sum_{k=1}^{K}\hat{r}_{i}^{t',H}(I, z|h'_{t',k})$$

(66)

where $norm' = \sum_{t'=1}^{t}\sum_{k=1}^{K}x_{t'}^{k}(I)$ denotes the normalizing factor, which is a positive constant for a certain memory $\tau_{R}^{i}$, and $h'_{t',k}$ is the termination state of the $k$-th sampled trajectory at iteration $t'$. In the second line of Equation (66), the second equality is valid based on the definition of sampled counterfactual regret (Equation (26) and (27)), which assigns non-zero values exclusively to information sets along the sampled trajectory. Now, we consider the expectation of $R_{i,*}^{t,H}(z|I)$ on the set of sampled trajectories $\{h'_{t',k}\}$:

$$\mathbb{E}_{\{h'_{t',k}\}}\left[R_{i,*}^{t,H}(z|I)\right] = \frac{1}{norm'}\sum_{t'=1}^{t}\sum_{k=1}^{K}\mathbb{E}_{h'_{t',k}}\left[\hat{r}_{i}^{t',H}(I, z|h'_{t',k})\right]$$

$$= \frac{1}{norm'}\sum_{t'=1}^{t}\sum_{k=1}^{K}r_{i}^{t',H}(I, z) = C_{1}R_{i}^{t,H}(z|I)$$

(67)

where $C_{1} = \frac{T}{K\times norm'}$ and the second equality holds due to Theorem 4. The second part of Proposition 2, i.e., $\mathbb{E}_{\{h'_{t',k}\}}\left[R_{i,*}^{t,L}(a|I, z)\right] = C_{2}R_{i}^{t,L}(a|I, z)$, can be demonstrated similarly.

## J    PROOF OF PROPOSITION 3

According to the definition of $\mathcal{L}_{\bar{\sigma},i}^{H}$ in Equation (10) and following the same process as Equation (65) - (66), we can obtain:

$$\bar{\sigma}_{i,*}^{T,H}(z|I) = \frac{1}{norm'}\sum_{t=1}^{T}\sum_{k=1}^{K}x_{t}^{k}(I)\sigma_{i}^{t,H}(z|I)$$

(68)

According to the law of large numbers, as $|\tau_{\bar{\sigma}}^{t,i}| \to \infty$ ($t \in \{1, \cdots, T\}$), we have:

$$\bar{\sigma}_{i,*}^{T,H}(z|I) \to \frac{\sum_{t=1}^{T}\pi^{q^{t,3-i}}(I)\sigma_{i}^{t,H}(z|I)}{\sum_{t=1}^{T}\pi^{q^{t,3-i}}(I)}$$

(69)

The equation above is based on the fact that, at a certain iteration $t$, the samples for updating the strategy of player $i$ are collected when $3 - i$ is the traverser, so the probability to visit a certain information set $I$ is $\pi^{q^{t,3-i}}(I)$. Ideally, to apply the law of large numbers, we should randomly select a single information set for each randomly-sampled trajectory and add its strategy distribution to the memory $\tau_{\bar{\sigma}}^{t,i}$. This guarantees that occurrences of information sets within each iteration $t$ are independent and identically distributed, as the sampling strategy within an iteration remains consistent. However, in practice (Algorithm 2), we gather strategy distributions of all information sets for the non-traverser along each sampled trajectory to enhance sample efficiency, which has been empirically proven to be effective.

To connect the convergence result in Equation (69) and the definition of $\bar{\sigma}_{i}^{T,H}$ in Equation (8), we need to show that $\forall I \in \mathcal{I}_{i}$, $t \in \{1, \cdots T-1\}$, $\frac{\pi^{q^{t,3-i}}(I)}{\pi^{q^{t+1,3-i}}(I)} = \frac{\pi_{i}^{\sigma^{t}}(I)}{\pi_{i}^{\sigma^{t+1}}(I)}$. According to the sampling scheme, $q_{p}^{t,3-i}$ is a uniformly random strategy when $p = 3 - i$, and it is equal to $\sigma_{p}^{t}$ when $p = i$. Therefore, we have:

$$\frac{\pi^{q^{t,3-i}}(I)}{\pi^{q^{t+1,3-i}}(I)} = \frac{\sum_{h\in I}\pi_{3-i}^{Unif}(h)\pi_{i}^{\sigma^{t}}(h)}{\sum_{h\in I}\pi_{3-i}^{Unif}(h)\pi_{i}^{\sigma^{t+1}}(h)} = \frac{\sum_{h\in I}\pi_{i}^{\sigma^{t}}(h)}{\sum_{h\in I}\pi_{i}^{\sigma^{t+1}}(h)} = \frac{\pi_{i}^{\sigma^{t}}(I)}{\pi_{i}^{\sigma^{t+1}}(I)}$$

(70)

It is satisfied in our/usual game settings that $\pi_{3-i}^{Unif}(h)$ remains consistent for all $h \in I$. This is attributable to the fact that histories within a single information set have the same height, and player $3 - i$ consistently employs a uniformly random strategy. Similarly, we can deduce that $\overline{\sigma}_{i,*}^{T,L}(a|I,z) \to \overline{\sigma}_i^{T,L}(a|I,z)$ using the aforementioned procedure.

## K  PROOF OF PROPOSITION 4

First, we present a lemma concerning the sampled baseline values $\hat{b}^{t+1}(h|h')$, as defined in Equation (12). This definition closely resembles that of the sampled counterfactual values in Equation (27), with two key distinctions: (1) $b^{t+1}$ is replaced with $b^t$, as $b^{t+1}$ is not yet available; and (2) $q^{t+1}$ is substituted with $q^t$, enabling the reuse of trajectories sampled with $q^t$ for updating $b^{t+1}$, thereby enhancing efficiency.

**Lemma 13.** *For all $h \in H$, $z \in Z(h)$, $a \in A(h)$, we have:*

$$\mathbb{E}_{h'}\left[\hat{b}^{t+1}(h|h')|h' \sqsupseteq h\right] = v^{t+1,H}(\sigma^{t+1}, h) \tag{71}$$

*Proof.* Given the similarity between $\hat{b}^{t+1}$ and $\hat{v}^{t+1,H}$, we can follow the proof of Lemma 4 and 5 to justify the lemma here.

$$E_{h'}\left[\hat{b}^{t+1}(h, z|h')|h' \sqsupseteq h\right]$$

$$= E_{h'}\left[\frac{\delta(hz \sqsubseteq h')}{q^{t,1}(z|h)}\left[\hat{b}^{t+1}(hz|h') - b^t(h, z)\right] + b^t(h, z)|h' \sqsupseteq h\right]$$

$$= P(hz \sqsubseteq h'|h' \sqsupseteq h)E_{h'}\left[\frac{1}{q^{t,1}(z|h)}\left[\hat{b}^{t+1}(hz|h') - b^t(h, z)\right] + b^t(h, z)|h' \sqsupseteq hz\right] + $$
$$P(hz \not\sqsubseteq h'|h' \sqsupseteq h)b^t(h, z) \tag{72}$$

$$= q^{t,1}(z|h)\left[\frac{1}{q^{t,1}(z|h)}\left[E_{h'}(\hat{b}^{t+1}(hz|h')|h' \sqsupseteq hz) - b^t(h, z)\right] + b^t(h, z)\right] + $$
$$(1 - q^{t,1}(z|h))b^t(h, z)$$

$$= E_{h'}\left[\hat{b}^{t+1}(hz|h')|h' \sqsupseteq hz\right]$$

According to Algorithm 2, the trajectories for updating $\hat{b}^{t+1}$ are sampled at iteration $t$ when player 1 is the traverser, so $P(hz \sqsubseteq h'|h' \sqsupseteq h) = q^{t,1}(z|h)$. Similarly, we can obtain $E_{h'}\left[\hat{b}^{t+1}(hz, a|h')|h' \sqsupseteq hz\right] = E_{h'}\left[\hat{b}^{t+1}(hza|h')|h' \sqsupseteq hza\right]$.

Next, we can employ these two equations to perform induction based on the height of $h$ within the game tree. If $h \in H_{TS}$, $E_{h'}\left[\hat{b}^{t+1}(h|h')|h' \sqsupseteq h\right] = \hat{b}^{t+1}(h|h') = u_1(h) = v^{t+1,H}(\sigma^{t+1}, h)$ based on the definition. If $hza \in H_{TS}$, we have:

$$E_{h'}\left[\hat{b}^{t+1}(h, z|h')|h' \sqsupseteq h\right] = E_{h'}\left[\hat{b}^{t+1}(hz|h')|h' \sqsupseteq hz\right]$$

$$= \sum_{a \in A(h)} \sigma_{P(h)}^{t+1,L}(a|h, z)E_{h'}\left[\hat{b}^{t+1}(hz, a|h')|h' \sqsupseteq hz\right]$$

$$= \sum_{a \in A(h)} \sigma_{P(h)}^{t+1,L}(a|h, z)E_{h'}\left[\hat{b}^{t+1}(hza|h')|h' \sqsupseteq hza\right] \tag{73}$$

$$= \sum_{a \in A(h)} \sigma_{P(h)}^{t+1,L}(a|h, z)v^{t+1,H}(\sigma^{t+1}, hza) = v^{t+1,L}(\sigma^{t+1}, hz)$$

Here, we employ the induction hypothesis in the fourth equivalence, and incorporate pertinent definitions for the remaining equivalences. It follows that:

$$E_{h'}\left[\hat{b}^{t+1}(h|h')|h' \sqsupseteq h\right] = \sum_{z \in Z(h)} \sigma_{P(h)}^{t+1,H}(z|h)E_{h'}\left[\hat{b}^{t+1}(h, z|h')|h' \sqsupseteq h\right]$$

$$= \sum_{z \in Z(h)} \sigma_{P(h)}^{t+1,H}(z|h)v^{t+1,L}(\sigma^{t+1}, hz) = v^{t+1,H}(\sigma^{t+1}, h) \tag{74}$$

By repeating the two equations above, we can show that $E_{h'}\left[\hat{b}^{t+1}(h|h')|h' \sqsupseteq h\right] = v^{t+1,H}(\sigma^{t+1}, h)$ holds for a general $h \notin H_{TS}$. $\qquad\square$

Next, we complete the proof of Proposition 4.

$$\mathcal{L}_b^{t+1} = \mathcal{L}(b^{t+1}) = \mathbb{E}_{h' \sim \tau_b^t}\left[\sum_{hza \sqsubseteq h'}(b^{t+1}(h, z, a) - \hat{b}^{t+1}(hza|h'))^2\right]$$

$$= \frac{\sum_{h'} N(h')\sum_{hza \sqsubseteq h'}(b^{t+1}(h, z, a) - \hat{b}^{t+1}(hza|h'))^2}{\sum_{h'} N(h')} \qquad (75)$$

Here, $N(h')$ denotes the number of occurrences of $h'$ in the memory $\tau_b^t$. Let $b^{t+1,*}$ denote a minimal point of $\mathcal{L}_b^{t+1}$. Utilizing the first-order necessary condition for optimality, we obtain: $\nabla\mathcal{L}(b^{t+1,*}) = 0$. Thus, for the $(h, z, a)$ entry of $b^{t+1,*}$, we deduce:

$$\frac{\partial \mathcal{L}(b^{t+1,*})}{\partial b^{t+1}(h, z, a)} = \frac{2\sum_{h' \sqsupseteq hza} N(h')(b^{t+1,*}(h, z, a) - \hat{b}^{t+1}(hza|h'))}{\sum_{h'} N(h')} = 0$$

$$b^{t+1,*}(h, z, a) = \frac{\sum_{h' \sqsupseteq hza} N(h')\hat{b}^{t+1}(hza|h')}{\sum_{h' \sqsupseteq hza} N(h')} \qquad (76)$$

The trajectories in $\tau_b^t$ can be considered as a sequence of independent and identically distributed random variables, since they are independently sampled with the same sample strategy $q^{t,1}$. Then, according to the law of large numbers, as $|\tau_b^t| \to \infty$, we conclude:

$$b^{t+1,*}(h, z, a) \to \frac{\sum_{h' \sqsupseteq hza} \pi^{q^{t,1}}(h')\hat{b}^{t+1}(hza|h')}{\sum_{h' \sqsupseteq hza} \pi^{q^{t,1}}(h')}$$

$$= \mathbb{E}_{h'}\left[\hat{b}^{t+1}(hza|h')|h' \sqsupseteq hza\right] = v^{t+1,H}(\sigma^{t+1}, hza) \qquad (77)$$

where the last equality comes from Lemma 13. It follows:

$$b^{t+1,*}(h, z) = \sum_a \sigma_{P(h)}^{t+1,L}(a|I(h), z)b^{t+1,*}(h, z, a)$$

$$\to \sum_a \sigma_{P(h)}^{t+1,L}(a|I(h), z)v^{t+1,H}(\sigma^{t+1}, hza) = v^{t+1,L}(\sigma^{t+1}, hz) \qquad (78)$$

## L  BENCHMARK INFORMATION

Table 3: Details of the Selected Benchmarks: For each benchmark, we provide its decision horizon, the number of nodes in the game tree, and the initial stack size for each player.

| Benchmark | Leduc | Leduc_10 | Leduc_15 | Leduc_20 | FHP | FHP_10 |
|---|---|---|---|---|---|---|
| Stack Size | 13 | 60 | 80 | 100 | 2000 | 4000 |
| Horizon | 4 | 20 | 30 | 40 | 8 | 20 |
| # of Nodes | 464 | 31814 | 67556 | 113954 | $2.58 \times 10^{12}$ | $3.17 \times 10^{13}$ |

Details of the benchmarks are summarized in Table 3.

## M  ABLATION STUDY RESULTS

HDCFR integrates the one-step option framework (Section 2.2) and variance-reduced Monte Carlo CFR (Section 3.2 and Appendix E). This section offers an ablation analysis highlighting each crucial

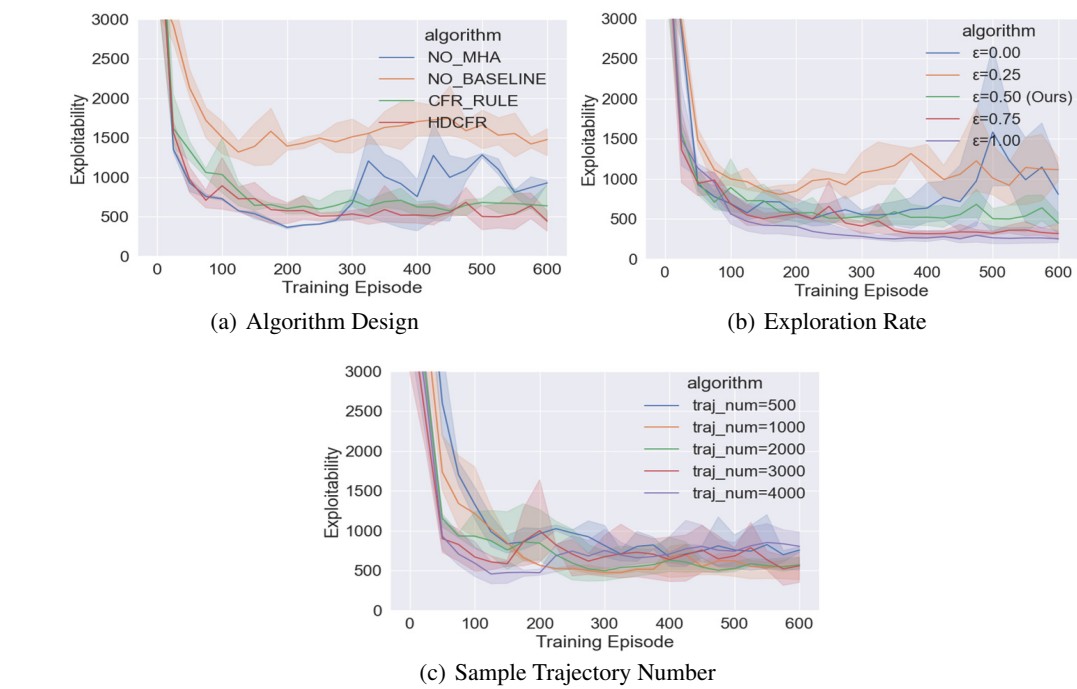

(a) Algorithm Design

(b) Exploration Rate

(c) Sample Trajectory Number

Figure 3: Learning curves of different ablations on Leduc_20. (a) Without the MHA component in the high-level strategy (NO_MHA) or the baseline function for variance reduction (NO_BASELINE), the learning performance degrades significantly. Following the CFR rule (Equation (7)) results in slightly slower convergence. (b) Increased randomness in the traverser's sample strategy enhances learning. (c) More sampled trajectories in each training episode boost initial convergence speed without affecting the convergent performance.

element of our algorithm: the option framework, variance reduction, Monte Carlo sampling, and CFR.

**The option framework:** The key component of the one-step option framework is the Multi-Head Attention (MHA) mechanism which enables the agent to temporarily extend skills and so form a hierarchical policy in the learning process. Without this component in the high-level strategy (i.e., NO_MHA in Figure 3(a)), the agent struggles to converge at the final stage, akin to the behavior observed for DREAM in Figure 1(d).

**Variance reduction:** In HDCFR, we incorporate a baseline function to reduce variance. This function proves pivotal for extended-horizon tasks where sampling variance can escalate. Excluding the baseline function from the learning process, as marked by NO_BASELINE in Figure 3(a), results in a substantial performance decline.

**Monte Carlo sampling:** During Monte Carlo sampling, as outlined in Section 3.3, the traverser should use a uniformly random sampling strategy. Yet, for fair comparisons with DREAM, we employ a weighted average of a uniformly random strategy (with the weight $\epsilon$) and the traverser's current strategy ($\sigma_i^t$). The controlling weight $\epsilon$ for HDCFR in the other experiments is set as 0.5, aligning with the configuration for the baselines. Figure 3(b) indicates that as $\epsilon$ increases, approximately there is a correlating rise in learning performance. Notably, our original design, i.e., setting $\epsilon = 1$, delivers the best result, amplifying the performance depicted in Figure 1(d). Another key aspect of Monte Carlo sampling is the number of sampled trajectories per training episode (i.e., the hyperparameter $K$ in Algorithm 1). According to Figure 3(c), increasing this count accelerates convergence during the initial training phase. However, it does not necessarily improve the final convergent performance and instead proportionally increases the overall training time.

**CFR:** As indicated by Brown et al. (2019) and Steinberger et al. (2020), slightly modifying the CFR updating rule (Equation (7)), that is, to greedily select the action with the largest regret rather than

use a random one when the sum $\mu^H, \mu^L \leq 0$, can speed up the convergence. We adopt the same trick and find that it can improve the convergence speed slightly, as compared to the original setting (i.e., CFR_RULE in Figure 3(a)).

# N    PSEUDO CODE OF HDCFR

---

**Algorithm 1** Hierarchical Deep Counterfactual Regret Minimization (HDCFR)

---

1: **Initialize** the counterfactual regret networks $R_{i,\theta}^{0,H}$, $R_{i,\theta}^{0,L}$, $\forall\, i \in \{1,2\}$ (collectively denoted as $R_\theta^0$), and the baseline network $b^1$, so that they return 0 for all inputs

2: **Initialize** the average strategy networks $\overline{\sigma}_{i,\phi}^{T,H}$, $\overline{\sigma}_{i,\phi}^{T,L}$, $\forall\, i \in \{1,2\}$ with random parameters

3: **Initialize** the replay buffer for the counterfactual regrets and average strategies, i.e., $\tau_R^i$, $\tau_{\overline{\sigma}}^i$, $\forall\, i \in \{1,2\}$ as empty sets

4: **for** $t = \{1, \cdots, T\}$ **do**

5:     **Initialize** the the replay buffer for the baseline function at iteration $t$: $\tau_b^t = \emptyset$

6:     **for** $i = \{1,2\}$ **do**

7:         **Define** the sample strategy profile at $t$ with $i$ being the traverser, i.e., $q^{t,i}$

8:         **for** traversal $k = \{1, \cdots, K\}$ **do**

9:             *HighRollout*$(\emptyset, R_\theta^{t-1}, \tau_R^i, \tau_{\overline{\sigma}}^{3-i}, \tau_b^t, q^{t,i}, b^t)$

10:         **end for**

11:     **end for**

12:     **for** $i = \{1,2\}$ **do**

13:         **Train** $R_{i,\theta}^{t,H}$, $R_{i,\theta}^{t,L}$ from scratch by minimizing Equation (9)

14:     **end for**

15:     $b^{t+1} = BaselineTraining(b^t, \tau_b^t, R_\theta^t, q^{t,1})$

16: **end for**

17: **for** $i = \{1,2\}$ **do**

18:     **Obtain** $\overline{\sigma}_{i,\phi}^{T,H}$, $\overline{\sigma}_{i,\phi}^{T,L}$ by minimizing Equation (10)

19: **end for**

20: **Return** $\{(\overline{\sigma}_{1,\phi}^{T,H}, \overline{\sigma}_{1,\phi}^{T,L}), (\overline{\sigma}_{2,\phi}^{T,H}, \overline{\sigma}_{2,\phi}^{T,L})\}$, i.e., the approximate Nash Equilibrium hierarchical strategy profile

21:

22: **function** *BaselineTraining*$(b^t, \tau_b^t, R_\theta^t, q^{t,1})$

23:     **for** $h'$ in $\tau_b^t$ **do**

24:         **for** $hza \sqsubseteq h'$ **do** (tracing back from $h'$ to its initial state)

25:             **Compute** $\hat{b}^{t+1}(hza|h')$ using $b^t$, $R_\theta^t$, and $q^{t,1}$, following Equation (12), where $R_\theta^t$ indicates $\sigma^{t+1}$ according to Equation (7)

26:         **end for**

27:     **end for**

28:     **Train** $b^{t+1}$ by minimizing Equation (11)

29:     **Return** $b^{t+1}$

30: **end function**

---

**Algorithm 2** Hierarchical Deep Counterfactual Regret Minimization (HDCFR) Continued

1: **function** *HighRollout*($h$, $R_\theta^{t-1}$, $\tau_R^i$, $\tau_{\bar\sigma}^{3-i}$, $\tau_b^t$, $q^{t,i}$, $b^t$)
2:     **if** $h \in H_{TS}$ **then**
3:         **Assign** $h' = h$
4:         **if** $i == 1$ **then**
5:             **Add** $h'$ to $\tau_b^t$
6:         **end if**
7:         **Return** $u_1(h')$
8:     **end if**
9:     $I = I(h)$, $p = P(h)$
10:     **Sample** an option $z \sim q^{t,i}(\cdot|h)$
11:     $\hat{v}^{t,L}(\sigma^t, hz|h') = LowRollout(h, z, R_\theta^{t-1}, \tau_R^i, \tau_{\bar\sigma}^{3-i}, \tau_b^t, q^{t,i}, b^t)$
12:     $\hat{v}^{t,H}(\sigma^t, h, z'|h') = b^t(h, z'), \forall\, z' \neq z$
13:     $\hat{v}^{t,H}(\sigma^t, h, z|h') = \frac{1}{q^{t,i}(z|h)} \left[ \hat{v}^{t,L}(\sigma^t, hz|h') - b^t(h, z) \right] + b^t(h, z)$
14:     $\hat{v}^{t,H}(\sigma^t, h|h') = \sum_{z \in Z(h)} \sigma_p^{t,H}(z|h) \hat{v}^{t,H}(\sigma^t, h, z|h')$
15:     **if** $p == i$ **then**
16:         $\hat{r}_i^{t,H}(I, \cdot|h') = (-1)^{i+1} \frac{\pi_{3-i}^{\sigma^t}(h)}{\pi^{q^{t,i}}(h)} \left[ \hat{v}^{t,H}(\sigma^t, h, \cdot|h') - \hat{v}^{t,H}(\sigma^t, h|h') \right]$
17:         **Add** $(I, t, \hat{r}_i^{t,H}(I, \cdot|h'))$ to $\tau_R^i$
18:     **else if** $p == 3 - i$ **then**
19:         **Compute** $\sigma_{3-i}^{t,H}(\cdot|I)$ based on $R_{3-i}^{t-1,H}(\cdot|I)$ following Equation (7)
20:         **Add** $(I, t, \sigma_{3-i}^{t,H}(\cdot|I))$ to $\tau_{\bar\sigma}^{3-i}$
21:     **end if**
22:     **Return** $\hat{v}^{t,H}(\sigma^t, h|h')$
23: **end function**
24:
25: **function** *LowRollout*($h$, $z$, $R_\theta^{t-1}$, $\tau_R^i$, $\tau_{\bar\sigma}^{3-i}$, $\tau_b^t$, $q^{t,i}$, $b^t$)
26:     $I = I(h)$, $p = P(h)$
27:     **Sample** an action $a \sim q^{t,i}(\cdot|h, z)$
28:     $\hat{v}^{t,H}(\sigma^t, hza|h') = HighRollout(hza, R_\theta^{t-1}, \tau_R^i, \tau_{\bar\sigma}^{3-i}, \tau_b^t, q^{t,i}, b^t)$
29:     $\hat{v}^{t,L}(\sigma^t, hz, a'|h') = b^t(h, z, a'), \forall\, a' \neq a$
30:     $\hat{v}^{t,L}(\sigma^t, hz, a|h') = \frac{1}{q^{t,i}(a|h,z)} \left[ \hat{v}^{t,H}(\sigma^t, hza|h') - b^t(h, z, a) \right] + b^t(h, z, a)$
31:     $\hat{v}^{t,L}(\sigma^t, hz|h') = \sum_{a \in A(h)} \sigma_p^{t,L}(a|h, z) \hat{v}^{t,L}(\sigma^t, hz, a|h')$
32:     **if** $p == i$ **then**
33:         $\hat{r}_i^{t,L}(Iz, \cdot|h') = (-1)^{i+1} \frac{\pi_{3-i}^{\sigma^t}(h)}{\pi^{q^{t,i}}(hz)} \left[ \hat{v}^{t,L}(\sigma^t, hz, \cdot|h') - \hat{v}^{t,L}(\sigma^t, hz|h') \right]$
34:         **Add** $(Iz, t, \hat{r}_i^{t,L}(Iz, \cdot|h'))$ to $\tau_R^i$
35:     **else if** $p == 3 - i$ **then**
36:         **Compute** $\sigma_{3-i}^{t,L}(\cdot|I, z)$ based on $R_{3-i}^{t-1,L}(\cdot|I, z)$ following Equation (7)
37:         **Add** $(Iz, t, \sigma_{3-i}^{t,L}(\cdot|I, z))$ to $\tau_{\bar\sigma}^{3-i}$
38:     **end if**
39:     **Return** $\hat{v}^{t,L}(\sigma^t, hz|h')$
40: **end function**

## O  THE USE OF LARGE LANGUAGE MODELS

Large Language Models (LLMs) were *not* used for the research idea, related-work selection, algorithm or experiment design, proofs or derivations, figure/table generation, or analysis/interpretation of results. We used LLMs only for lightly improving grammar and phrasing on a small number of sentences. No technical content was generated by LLMs; All suggestions were reviewed and, when appropriate, rewritten by the authors. No confidential or proprietary data were provided to LLMs.The authors take full responsibility for the contents of this paper, and LLMs are not authors.

