# OpenReview forum: "Hierarchical Deep Counterfactual Regret Minimization"
_ICLR.cc/2026/Conference — ICLR 2026 Conference Withdrawn Submission_

### Official Review · Reviewer_oGc7 · 2025-10-14

**Soundness:** 1
**Presentation:** 1
**Contribution:** 1
**Rating:** 0
**Confidence:** 5

**Summary:**

This paper proposes an incremental approach combining hierarchical RL and Deep CFR, achieving limited performance gains under constrained conditions compared to a baseline from seven years prior. However, the article neglects to compare with several demonstrably superior baselines.

**Strengths:**

1. The paper demonstrates strong reproducibility, proposes a relatively straightforward algorithm, and provides implementation code.

2. The paper presents an approach for interpreting CFR-based policies in a manner readily comprehensible to humans.

3. The thesis represents a substantial amount of work and provides detailed theoretical proofs.

4. This paper presents the first approach to combine hierarchical RL with CFR.

**Weaknesses:**

Main Concern: During the previous round of peer review, reviewers explicitly noted that the baseline DREAM (Steinberger et al. 2020) method compared in this paper is not SoTA. Escher (McAleer et al. 2023) demonstrates significant improvements over DREAM across multiple benchmarks and represents the SoTA CFR-based model-free reinforcement learning method. The authors acknowledged in the prior review that they were unfamiliar with the Escher method and committed to incorporating comparisons with Escher in the revised version. However, at this ICLR conference, the authors have once again omitted Escher, failing to mention the method even once throughout the entire paper. They further mislead readers by erroneously asserting that DREAM is the current state-of-the-art method (line 673). On this basis, I contend that this paper contains fundamental factual errors and should not be published at any conference, let alone a premier conference such as ICLR. Moreover, Escher was published at ICLR.


Other Concerns:

1.Unclear motivation: When addressing high-depth challenges, the depth-limited search framework (Zhang & Sandholm 2025; Anonymous, 2026) can perform deep search and reasoning, yielding superior results compared to RL-based approaches. The advantages of Deep CFR and other RL-based methods are fasting training speed and proving abstraction schemes or value estimations for the large game. However, the training speed of HDCFR is similar to previous methods. More importantly, HDCFR is a hierarchical version CFR, that we cannot get the abstraction schemes or value estimations in the large game due to there may be multiple skill selections beforehand, this exponential number of choices will lead to different HDCFR states. Moreover, Deep CFR can handle scenarios with large-scale action spaces, whereas HDCFR requires establishing a feasible/unfeasible representation for each action. This implies that HDCFR appears incapable of processing scenarios involving large-scale action spaces. Finally, HDCFR can only support inefficient Outcome Sampling, whereas Deep CFR itself can support more efficient External Sampling. In summary, even when compared solely to Deep CFR methods from seven years ago, HDCFR offers limited improvements while imposing numerous constraints.

2.The paper mentioned HDCFR could potentially mirror more human-like decision-making, however, I cannot see any evidence in the paper that HDCFR strategy is similar to human strategy. HDCFR's game strategy is determined based on the current information set; in contrast, Preference-CFR (Ju et al. 2025) enables the use of macro-level game strategies under conditions of limited exploitability, which more closely modeling human player strategy. Furthermore, Preference-CFR intuitively demonstrates the gap between human strategy and the Nash equilibrium, whereas HDCFR fails to illustrate the specific practical guidance offered by the chosen strategy within the game.

3.The paper introduced a new variant of CFR named HCFR. The paper provided a theoretic analysis of HCFR, however, the regret bound is $|Z_i|$ times larger than CFR. More importantly, the paper do not show the performance of HCFR in the Tabular setting against other CFR variants. Therefore, HCFR holds no practical significance either in theory or in practice. Moreover, HDCFR employs skill transfer, whereby the current skill is selected based on the current information set and the previous skill choice; whereas HCFR appears to select the current skill directly based on the information set. This implies that the theoretical framework of HCFR cannot be directly applied to HDCFR, meaning the theoretical underpinnings of HDCFR still require further substantiation.

4.The standard of this paper is rather low. Its introduction is excessively brief, leaving the reader unable to grasp the purpose of HDCFR. The formulas within the main text are overly redundant, making it tedious to read and difficult to identify the key points. There is a lack of a diagram to visually represent the HDCFR algorithm, while the experimental graphs occupy an excessive amount of space. The main body does not fill the full nine pages.

5.The experiments presented herein are unconvincing. They were conducted solely on poker variants with restricted actions and rules, failing to extend to conventional poker benchmarks such as no-limit Hold'em. This contradicts the paper's stated objective of addressing large IIG. Furthermore, as a general-purpose algorithm, it warrants further experimentation on different games. Moreover, across all experiments, the performance gap between HDCFR and DREAM is negligible during the early training stages. For Deep CFR-class algorithms, the early training phase is of particular concern, as their inherent potential is limited. If sufficient improvement cannot be achieved during this critical period, it becomes even more impossible to compete with the search frameworks.


Minor: The abstract in OpenReview does not match the abstract in the paper.


Eric Steinberger, Adam Lerer, Noam Brown. DREAM: deep regret minimization with advantage Baselines and model-free learning. Arxiv 2020.

Stephen Marcus McAleer, Gabriele Farina, Marc Lanctot, Tuomas Sandholm. ESCHER: Eschewing Importance Sampling in Games by Computing a History Value Function to Estimate Regret. ICLR 2023.

Qi Ju, Thomas Tellier, Meng Sun, Zhemei Fang, YunFeng Luo. Preference-CFR: Beyond Nash Equilibrium for Better Game Strategies. ICML 2025.

Brian Hu Zhang, Tuomas Sandholm. General search techniques without common knowledge for imperfect-information games, and application to superhuman Fog of War chess. Arxiv 2025.

Anonymous. Look-ahead Reasoning with a Learned Model in Imperfect Information Games. ICLR 2026 submission.

**Questions:**

1. Were you aware of the Escher method at the time of submission?

2. When we wish to ascertain the actual strategy of an information set, how might we expeditiously determine this through the HDCFR? I think the HDCFR can only ascertain the policy of an information set under a specific sequence of techniques, but cannot rapidly compute the policy of an information set without considering the sequence of techniques.

---

### Official Review · Reviewer_2XHu · 2025-10-29

**Soundness:** 1
**Presentation:** 2
**Contribution:** 2
**Rating:** 2
**Confidence:** 4

**Summary:**

This paper proposes Hierarchical Deep Counterfactual Regret Minimization (HDCFR), a hierarchical extension of Deep CFR for solving imperfect-information games with large state spaces and long decision horizons. The method integrates the one-step option framework with variance-reduced Monte Carlo CFR, allowing agents to learn high-level skill-selection policies and low-level action policies simultaneously. Theoretical results establish convergence guarantees in the tabular case, and the authors extend these to the deep learning setting through function approximation. Empirical evaluations on benchmark games such as Leduc Hold’em and Flop Hold’em Poker demonstrate the potential advantages of HDCFR over existing baselines like DREAM and NFSP.

**Strengths:**

The paper presents a clear and meaningful innovation by extending Deep CFR into a hierarchical setting. The theoretical development appears solid and well-grounded, providing formal convergence guarantees and consistent formulations between the tabular and neural approximations. The idea of combining option-based temporal abstraction with counterfactual regret minimization is novel and potentially impactful, especially for long-horizon imperfect-information games.

**Weaknesses:**

## Writing and Organization

Since Section 2 already provides a BACKGROUND, introducing another subsection 3.1 PRELIMINARIES feels redundant. In my view, most of Section 3 belongs to the algorithm construction part and could be merged into the current 3.2. Fundamental materials such as Theorem 1 and Equation (5) would be more appropriate in the BACKGROUND section. This restructuring would make the paper’s logical flow clearer and the transitions between sections more natural.


---

## Experiments

While the theoretical contribution and the core idea are valuable, the experimental presentation significantly weakens the overall impression of the paper.

First, all figures show noticeable stretching artifacts, which look unprofessional. Likewise, Table 1 should use a uniform numeric precision (two decimal places).

Second, using iteration count on the x-axis is not ideal because each algorithm has very different computational costs per iteration, making direct comparison meaningless. It would be more informative to use training time or the number of traversed nodes as the x-axis. Moreover, plotting two different y-axes in a single figure is confusing; using a single axis—possibly with a logarithmic scale—would improve clarity.

Third, I have concerns about the reliability of the reported results. The algorithm used in the comparative experiment does not appear to be optimized.
- In the Leduc_15 setting, the performance of NFSP keeps deteriorating during training and even becomes much worse than its initial policy. This is highly suspicious and likely caused by an implementation bug or inappropriate hyper-parameters; Or if you have already carefully adjusted the parameters and ensured that there are no bugs, you should also clarify this fact in the main text.
- In addition, the exploitability curve of DREAM in your paper exhibits a “decrease-then-increase” pattern that does not appear in the original DREAM experiments for Leduc or FHP. You should clarify this discrepancy or tune parameters (e.g., learning rate) to eliminate the instability.
- Furthermore, the paper does not disclose the hyper-parameter settings used for the compared baselines. If those baselines were not run with their optimal configurations, the comparison is not meaningful. Please include a complete parameter table in the main text or appendix.


Finally, it is surprising that the paper provides almost no description of the neural-network architecture. It only briefly mentions the use of an MHA module, without explaining the detailed design. Given that combining multi-head attention with poker-style games appears novel and potentially impactful, this aspect deserves a much more thorough discussion and justification. If it is more effective than previous MLP, LSTM, or CNN methods, it could even become the core contribution of this paper.


---

## Other Issues

The provided code link has already expired. Please update it to a valid repository and ensure that it remains accessible throughout the review period.

**Questions:**

Refer to the previous section

---

### Official Review · Reviewer_J9zn · 2025-10-30

**Soundness:** 3
**Presentation:** 2
**Contribution:** 2
**Rating:** 4
**Confidence:** 3

**Summary:**

This paper proposes HDCFR, a hierarchical version of Deep CFR, to improve learning efficiency in large IIGs, especially under long decision horizons. The method decomposes strategies into high-level options and low-level primitive actions, inspired by the option framework in RL. The authors develop both a tabular version with theoretical convergence guarantees and a neural version that supports model-free, simulator-only settings via outcome sampling and variance reduction. Extensive experiments on Leduc variants and FHP show HDCFR achieves lower exploitability and stronger head-to-head performance than baselines like DREAM, OSSDCFR, and NFSP. The method also enables skill reuse and transfer across similar tasks.

**Strengths:**

1. This paper proposes the first hierarchical Deep CFR framework with a principled design and convergence guarantee. They also introduce novel variance-reduction techniques for model-free outcome sampling in hierarchical CFR.
2. Theoretical derivations are solid, and the neural version retains fidelity to the tabular regret updates.
3. Experimental results show that HDCFR outperforms baselines in IIGs. They also demonstrate skill transfer across tasks and analyze hierarchical strategies in a meaningful way.

**Weaknesses:**

1. The method introduces multiple moving parts. Although theoretically sound, practical training complexity is relatively high.
2. The method is only evaluated on poker variants, actually not very large. Generalization to other types of large-scale IIGs is not explored.
3. Notation and explanation of the hierarchical regret estimators and baseline updates are dense and could be more intuitive for readers outside this subfield.

**Questions:**

1. The theoretical analysis is clean in the tabular case. In the deep setting, the function approximators may introduce approximation error and instability. How sensitive is the method to such errors?
2. How is the number of options (skills) determined in practice? Is it manually fixed, heuristically chosen, or learned? Can the suboptimal choice of this number degrade performance?
3. The paper assumes a shared intra-option action space $A(h)$ regardless of the selected skill. Would it be possible or beneficial to allow different skills to constrain or specialize the action space?

---

### Official Review · Reviewer_wMmW · 2025-10-31

**Soundness:** 3
**Presentation:** 3
**Contribution:** 3
**Rating:** 4
**Confidence:** 4

**Summary:**

This paper introduces Hierarchical Deep Counterfactual Regret Minimization (HDCFR), a hierarchical extension of Deep CFR designed to learn temporally abstract strategies in imperfect-information games (IIGs).  The method incorporates the *option framework* from hierarchical reinforcement learning into the CFR paradigm, enabling skill-based decomposition of decision-making.

The authors first establish a tabular theoretical framework, defining hierarchical regret and proving convergence at the same asymptotic rate $O(T^{-0.5})$ as standard CFR.  They then extend this to large-scale, model-free environments using neural networks for regret and policy approximation, along with a ****variance-reduced Monte Carlo sampling scheme.**

Empirically, HDCFR outperforms baseline methods such as DREAM and NFSP on Leduc Poker and Flop Hold’em Poker, and demonstrates partial skill transferability between related tasks.

**Strengths:**

S1: **Solid theoretical basis**. Extends CFR to hierarchical decision spaces while retaining convergence.

S2: **Variance reduction**. Introduces a practical baseline to stabilize regret estimation.

S3: **Empirical performance**. Achieves lower exploitability than DREAM and NFSP in poker domains.

S4: **Skill reuse**.Demonstrates that low-level skills can transfer across related games.

S5: **Interpretability**. The hierarchical structure provides insight into the decomposition of long-term strategy.

**Weaknesses:**

**W1: Limited theoretical novelty**

The paper’s theoretical contribution is primarily structural rather than conceptual. It extends the classical CFR framework by introducing a hierarchical policy structure, replacing primitive actions \(a\) with hierarchical pairs \((z, a)\).  However, the regret formulation, proof technique, and convergence bound remain identical to the original CFR framework. The work does not present a new equilibrium notion, regret-bound improvement, or hierarchical-specific convergence analysis. While the idea of hierarchical abstraction is interesting, the theoretical depth and originality are limited.

---

**W2: Simplifying independence assumptions**

The hierarchical design assumes conditional independence between options across timesteps (e.g., \(z_t \perp z_{1:(t-2)} | z_{t-1}\)).  This assumption simplifies derivations but ignores longer-term dependencies that are often crucial in hierarchical or multi-stage imperfect-information games.  As a result, the framework may not capture the temporal coupling between strategic choices over extended horizons.  Relaxing this assumption or analyzing its effect on convergence would strengthen the theoretical foundation.

---

**W3: Missing link to the classical CFR formulation**

Although HDCFR claims to generalize CFR to hierarchical settings, it does not explicitly prove that the framework reduces to standard CFR when the hierarchy collapses (e.g., when \(|Z| = 1\)).  Such a reduction is important to establish mathematical consistency with existing CFR theory.  Without it, it remains unclear whether HDCFR is a strict generalization or simply a reparameterization of the same regret structure.

---

**W4: No convergence analysis for the deep version**

The convergence guarantees are only established in the **tabular setting**.  The deep variant, which relies on neural approximations of regret and strategy functions, lacks any theoretical analysis of approximation error or stability.  This is a critical gap, as deep approximations can break regret guarantees through sampling noise and non-stationary updates. Providing an approximate-convergence or bounded-regret analysis for the deep setting would make the method more convincing.

---

**W5: Static and predefined hierarchy**

The hierarchical structure—namely, the set of options or skills available at each decision point—is **predefined** and remains fixed throughout training. The model does not include any mechanism for automatically discovering, merging, or refining these options based on the data or learning dynamics. As a result, the hierarchy cannot adapt to the complexity of different games or tasks, which limits the flexibility and generality of the proposed approach. Introducing a data-driven or regret-guided way to adjust the hierarchy during learning could make the framework more adaptive and scalable.

---

**W6: Missing comparison with recent work**

The paper does not include comparisons or discussions of more recent developments in deep CFR or related counterfactual regret-minimization methods.  Without such context, it isn't easy to assess how the proposed hierarchical extension advances beyond the current state of the art.

**Questions:**

Q1. Could the authors formally clarify whether HDCFR reduces exactly to classical CFR when the hierarchy collapses (i.e., when \(|Z| = 1\))? This reduction would help confirm the mathematical consistency of the framework.

Q2. The paper assumes conditional independence between hierarchical options $(z_t \perp z_{1:(t-2)} | z_{t-1})$.  How strong is this assumption in practice, and how would performance or convergence change if dependencies among options were introduced?

Q3. The convergence analysis is presented only for the tabular setting. Have the authors considered providing an approximate or empirical convergence argument for the deep variant to address potential approximation bias?

Q4. The hierarchy (option set \(Z(h)\)) appears to be manually specified. Could the method be extended to learn or adapt the hierarchy automatically during training, perhaps based on regret magnitude or clustering of state–action pairs?

Q5. The paper compares mainly with earlier baselines such as Deep CFR and NFSP. Have the authors evaluated or discussed more recent variants of the CFR or deep regret minimization methods? Including such a comparison would help clarify how the proposed approach advances beyond existing work.

---

### Note · Authors · 2025-11-12

I have read and agree with the venue's withdrawal policy on behalf of myself and my co-authors.